


# ArcticBeach v1.0: A physics-based parameterization of pan-Arctic coastline erosion

Rebecca Rolph[1,2], Pier Paul Overduin[1], Thomas Ravens[3], Hugues Lantuit[1,4], and Moritz Langer[1,2]

[1]Alfred Wegener Institute Helmholtz Centre for Polar and Marine Research, Telegrafenberg A45, 14473, Potsdam, Germany
[2]Geography Department, Humboldt-Universität zu Berlin, Unter den Linden 6, 10099, Berlin, Germany
[3]University of Alaska Anchorage, 3211 Providence Dr., Anchorage, AK 99508, United States
[4]University of Potsdam, Am Neuen Palais 10, 14469, Potsdam, Germany

**Correspondence:** Rebecca J. Rolph (rebecca.rolph@awi.de)

**Abstract.** In the Arctic, air temperatures are warming and sea ice is declining, resulting in larger waves and a longer open water season, all of which intensify the thaw and erosion of ice-rich coasts. This change in climate has been shown to increase the rate of Arctic coastal erosion, causing problems for industrial, military, and civil infrastructure as well as changes in nearshore biogeochemistry. Numerical models that reproduce historical and project future Arctic erosion rates are necessary to understand

how further climate change will affect these problems, and no such model yet exists to simulate the physics of erosion on a pan-Arctic scale. We have coupled a bathystrophic storm surge model to a simplified physical erosion model of a partially frozen cliff and beach. This Arctic erosion model, called ArcticBeach v1.0, is a first step toward a parameterization of Arctic shoreline erosion for larger-scale models, which are not able to resolve the fine spatial scale (up to about 40 m) needed to capture shoreline erosion rates from years to decades. It is forced by wind speeds and directions, wave period and height, sea

surface temperature, all of which are masked during times of sea ice cover near the coastline. Model tuning requires observed historical retreat rates (at least one value), as well as rough nearshore bathymetry. These parameters are already available on a pan-Arctic scale. The model is validated at two study sites at Drew Point (DP), Alaska, and Mamontovy Khayata (MK), Siberia, which are respectively located in the Beaufort and Laptev Seas, on different sides of the Arctic Ocean. Simulated cumulative retreat rates for DP and MK respectively (169 and 170 m) over the time periods studied at each site (2007 - 2016, and 1995 -

2018) are found to be within the same order of magnitude as observed cumulative retreat rates (172 and 120 m). Given the large differences in geomorphology and weather systems between the two study sites, this study provides a proof-of-concept that ArcticBeach v1.0 can be applied on very different partially frozen coastlines. ArcticBeach v1.0 provides a promising starting point to project the retreat of Arctic shorelines, or to evaluate historical retreat in places that have had few observations. Further, this model can provide estimates of the flux of sediment from land to sea for Arctic nearshore biogeochemical studies, while

leaving an opportunity for further development of modelling the physics of a partially frozen shoreline.

## 1 Introduction

Arctic coastlines are increasingly vulnerable to erosion due to warmer temperatures (Biskaborn et al., 2019) through the destabilization of frozen cliffs, reduced sea ice protection from bigger waves (Casas-Prat and Wang, 2020; Overeem et al.,





2011), especially as freeze-up becomes delayed further into the fall storm season. Large-scale atmospheric patterns have
been recently attributed to driving the variability of Arctic shoreline erosion (Nielsen et al., 2020) and statistical methods
might therefore show promising results to simulate erosion rates. However, understanding the most important root causes of
Arctic shoreline change can be only gained through careful evaluation of the physical processes involved. Although extensive
process-based models exist (Bull et al., 2020; Ravens et al., 2017, 2012; Hoque and Pollard, 2009; Barnhart et al., 2014)
these have only been designed for very specific stretches of coastline and mostly focused on the quickly eroding Drew Point
and greater southern Beaufort coastline. These models require extremely detailed initialization data and only pertain to their
respective stretch of coastline. These types of models are thus not designed for use on a pan-Arctic level where detailed data
on geomorphological characteristics and bathymetry are not available. In addition, notch erosion (undercutting of a steep bluff
by water or waves) is a key aspect in their formulation of the coastline retreat process. While this process is important in some
locations along the Arctic, notch erosion does not apply on a pan-Arctic scale (Lantuit et al., 2012). Further, most existing
erosion models are computationally expensive and require long run times, not suitable for efficient physical modelling on
pan-Arctic erosion scale. Therefore, the need remains to form a physics-based numerical model that can be applied across all
partially frozen shorelines. We present, for the first time, a general numerical erosion model that can serve as a starting point
for a physics-based parameterization of Arctic shoreline erosion in earth system models.

The processes involved in Arctic shoreline erosion are different than their mid- and low- latitude counterparts due to the
cold temperatures and presence of ice and frozen soils. Shorelines along the Arctic can be frozen and connected to landfast
sea ice (Mahoney, 2018), protecting the bluffs and beaches from abrasive wave action. However, strong winds and storm
surges can also push ice roughly onto shore, causing erosion, debris influx, and significant destruction of infrastructure and
cultural sites (Bogardus et al., 2020). During the summer, the open water period allows for relatively warmer water to thaw the
submerged part of the beach, and warmer air temperatures to thaw the exposed part of the shoreline. Thawing shorelines are
especially vulnerable to erosion (Aré, 1988), and climate change accelerates this process due to the lengthening open water
season and higher sea and air surface temperatures (Barnhart et al., 2014). Social and economic costs of erosion are high,
with entire villages having to relocate (Albert et al., 2018; Hamilton et al., 2016). Nearshore biogeochemistry is also heavily
impacted by nutrient-laden sediment supplied into the Arctic Ocean, with roughly one third of the Arctic Ocean primary
production supported by riverine and coastal sediment inputs (Terhaar et al., 2021). Further, thawing and eroding coastlines
can exacerbate climate change by releasing previously sequestered carbon from the soil into the atmosphere (Vonk et al., 2012;
Fritz et al., 2017).

The paper set-up is defined as follows. In Section 2, we describe the erosion model and the physical mechanisms and
associated initialization parameters included for simulating the erosion of a partially frozen cliff and beach. Next, we describe
the water level model, and how it uses wind forcing to generate a time history of relative water levels at the coastline, which
are then used to drive the erosion model. Data used for the validation of both the erosion and storm surge model components
are also provided. In Section 3, model results and validation are given, along with model sensitivity to critical parameters.
Sections 4 and 5 provide a discussion of the results and conclusions.



## 2 Methods

We have coupled the framework of an existing 1-D Arctic coastline erosion model (Kobayashi et al., 1999) with a bathystrophic
storm surge model (Freeman et al., 1957), forced by wind speed and direction, and initialized using existing bathymetric in-
formation of our study sites. The idealized set-up of the erosion model (Figure 1) includes a beach and cliff profile, assuming
uniform conditions alongshore. Conceptually, the model simulates thawing of the beach and cliff sediments according to con-
vective heat transfer controlled by water level and temperature. Thawed material is assumed to be prone to erosion depending
on water level and wave action. The process of mass transfer is simulated by emulating a cascade of cliff erosion, beach depo-
sition, and beach erosion. According to the resulting mass balance, the beach and cliff profiles are adjusted assuming constant
beach and cliff inclination. Small scale processes such as niche formation are neglected in this coarse-scale approach. Further
description of the beach and cliff model parameters are given in Section 2.1.

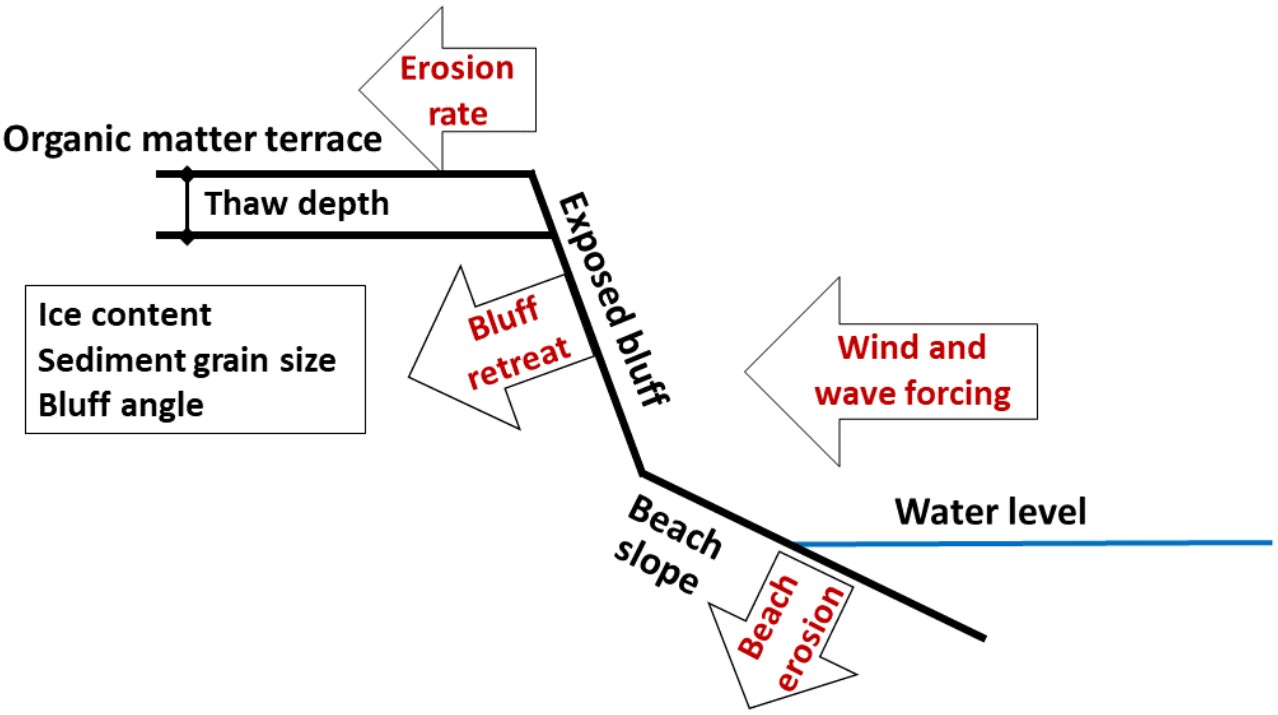

**Figure 1.** Model sketch illustrating basic physical model parameters (black) and processes (red). Wind forcing, masked during times of sea
ice cover, is taken from the ERA-Interim reanalysis (Dee et al., 2011) dataset to force a coupled storm surge model. This provides water
level data to the erosion model, driving the bluff retreat and beach erosion through a heat and volume balance. Sea surface temperature, wave
height, and wave period are also taken into account, as well as the prescribed cliff and beach parameters of volumetric ice content, sediment
grain size, cliff height, thaw depth, and cliff and beach angle.





## 2.1 Erosion model

The erosion model used in this study is constructed from heat and sediment volume balances in order to predict horizontal cliff

retreat and vertical erosion of a fronting beach. A full description of the framework for this model can be found in Kobayashi et al. (1999), but we provide a brief overview of the main driving mechanisms here. Wave action and water levels drive convective heat transfer, and thaw ice-bonded sediments comprising the cliff and beach. When cliff sediment, with its initially prescribed coarse sediment fraction, is released via melting ice between grains of sediment, this coarse sediment is deposited onto the beach, while the remaining fraction of cliff sediment (the fine sediment) is assumed to be transported offshore by

the seawater. The amount of coarse sediment (defined by a grain size threshold) that remains on the beach is determined by a volume balance. The volume balance is defined as follows: the rate of coarse sediment transport transported away from the beach cannot exceed a so-called potential sediment transport rate that is determined largely by the beach angle and water level. In general, steeper beach angles and higher water levels lead to higher potential coarse sediment transport rates away from the beach and offshore. Flat beaches and low water levels will result in a low amount of coarse sediment that could be

transported offshore. More detail of modelled mechanisms driving cliff and beach erosion are given in the subsequent sections (Sections 2.1.1 and 2.1.2) and also in Kobayashi et al. (1999).

### 2.1.1 Cliff erosion

The rate of the cliff retreat is determined by the heat transfer into the exposed frozen cliff face assuming isothermal frozen sediments at freezing temperature ($0°C$). The rate of cliff retreat ($\frac{\partial R}{\partial t}$) is, thus, defined by the rate of melting of interstitial ice

and subsequent release of cliff sediment determined by the energy supplied divided by the energy required to thaw the part of the cliff face that is exposed to seawater. This expression is given by

$$\frac{\partial R}{\partial t} = \frac{l_c h_c (T_w - T_m)}{L_c (H - B_c)} \: for \: d_c > 0, \tag{1}$$

where $H$ is the cliff height [m], $d_c$ is the depth of the water level at the cliff toe [m], $l_c$ is the length of the cliff exposed to the

water [m], $h_c$ is the convective heat transfer coefficient [J/(s m$^2$ °C] with a volumetric latent heat of fusion $L_c$ [J/m$^3$], $B_c$ is the initial thaw depth on top of the cliff [m], and the temperature of the seawater $T_w$ [°C]. The volume of cliff coarse sediment, per unit width and unit horizontal length, is given by

$$P_c B_c + v_c (H - B_c), \tag{2}$$

where $P_c$ is the coarse sediment volume per unit volume of unfrozen cliff sediment [m$^3$/m$^3$], and $v_c$ is the coarse sediment volume per unit volume of the frozen cliff sediment [m$^3$/m$^3$]. The rate of the coarse sediment supplied to the fronting beach





is assumed equal to the offshore coarse sediment transport rate per unit width at the cliff toe. Note that this does not allow for accumulation of sediment directly at the base of the cliff. The sediment supply from the eroding cliff (assumed to be zero if water does not reach the cliff), is taken into account when calculating the rate of vertical beach erosion and sediment transported from the beach offshore.

### 2.1.2 Beach erosion

The potential coarse beach sediment transport rate (i.e. sediment transport from the beach towards offshore) mentioned in Section 2.1 is estimated using available empirical formulas for cross-shore sediment transport on ice-free sandy beaches (Kriebel and Dean, 1985) and adjusted to accommodate a coarse sediment fraction (Kobayashi, 1987). The potential rate of beach sediment is the upper limit for the rate of transport of sediment away from the beach. When the actual sediment transport rate supplied to the beach from the retreating cliff exceeds the potential beach sediment transport rate, then coarse sediment is allowed to accumulate on the beach. However, if insufficient sediment is supplied by the cliff to the beach to accommodate a greater potential transport away from the beach, then no sediment will accumulate on the beach. The balance of both of these processes controls the change in unfrozen coarse sediment on the beach. The change in thickness of unfrozen coarse sediment on the beach is not only determined by the actual transport rate away from the beach and the sediment supply onto the beach from the cliff, but also is influenced by the release of sediment from thawing the beach itself. If the cliff is not providing enough sediment to keep up with the sediment being transported away by the seawater, then the frozen beach is exposed to thaw by the seawater. This results in vertical beach thaw rate defined as $\frac{\partial D}{\partial t}$, as given by

$$\frac{\partial D}{\partial t} = \frac{h_{\mathrm{b}}(T_{\mathrm{w}} - T_{\mathrm{m}})}{L_{\mathrm{b}}},$$
(3)

where $h_{\mathrm{b}}$ is the convective heat transfer coefficient on the exposed frozen beach sediment [J/(s m$^2$ °C], $T_{\mathrm{w}}$ is the temperature of the seawater [°C], $T_{\mathrm{m}}$ is the melting point of the interstitial ice between the sediment grains (which can be adjusted for salinity) [°C], and $L_{\mathrm{b}}$ is the volumetric latent heat of fusion [J/m$^3$]. As long as there is coarse sediment available on top of the frozen part of the beach, the beach is assumed to be protected from thaw and vertical beach erosion does not occur ($\frac{\partial D}{\partial t} = 0$).

To summarize, the change in thickness of unfrozen coarse sediment on the beach is determined by a sediment volume balance controlled by the three major sediment fluxes: (i) the potential offshore beach sediment transport largely determined by beach angle and water level, (ii) cliff sediment supply onto the beach, and (iii) the release of previously-frozen beach sediment now available for offshore transport due to an increase in beach thaw depth. The change in thickness of unfrozen coarse sediment on the beach, $\frac{\partial B}{\partial t} = 0$, is given by

$$\frac{\partial B}{\partial t} = \frac{q_{\mathrm{c}} + q_{\mathrm{melt}} - q_{\mathrm{b}}}{P_{\mathrm{b}} W},$$
(4)





where $q_c$ is the coarse sediment supply rate from the eroding cliff [m$^2$/s] (volume of cliff coarse sediment from Equation 2 times rate of cliff retreat from Equation 1), $q_{melt}$ is the coarse sediment supply rate due to beach thaw [m$^2$/s] over beach width $W$ [m], $q_b$ is the offshore transport rate of unfrozen coarse sediment at the offshore model boundary [m$^2$/s], and $P_b$ is the coarse sediment volume per unit volume of frozen beach sediment.

### 2.1.3 Cliff and beach parameters

The cliff and beach are each initialized with values for slope, coarse sediment fraction per unit volume for each unfrozen and frozen sediment, sand roughness length (from grain size) and initial thaw depth. The beach width and cliff height are also specified at the start of the model run. Default values and reasonable ranges for many of these parameters, taken from referenced literature, were tested in a sensitivity analysis (see Section 2.6) for both of our chosen study sites, Mamontovy Khayata (MK), Bykovsky Peninsula, Siberia and Drew Point (DP), Alaska, USA (Figure 2). These values, their ranges, and associated references are given in Table 1.

| Parameter | Low | Default | High | Reference |
|---|---|---|---|---|
| Initial unfrozen beach sediment thickness [m] | 0.5 | 1 | 2 | Kobayashi et al. (1999) |
| Cliff height [m] | 5 (MK), 1 (DP) | 10 (MK), 3 (DP) | 20 (MK), 10 (DP) | Overduin et al. (2007), Jones et al. (2018) |
| Cliff angle [degrees] | 45 | 60 | 90 | Overduin et al. (2007), Jones et al. (2018) |
| Initial unfrozen cliff sediment thickness [m] | 0.1 | 0.2 | 0.5 | Günther et al. (2015) |
| Coarse sediment volume per unit volume unfrozen cliff sediment [%] | 5 | 10 | 20 | Kobayashi et al. (1999), Overduin et al. (2014) |
| Ice volume per unit volume frozen cliff sediment [%] | 60 | 80 | 90 | Overduin et al. (2007), Kanevskiy et al. (2013) |

**Table 1.** Parameters used in the Monte-Carlo sensitivity studies to initialize the erosion model are given as 'low', 'default' and 'high' values.





## 2.2 Bathystrophic storm surge model

Due to the extremely limited number of tide gauges spaced across Arctic coastlines, we provide water level to our erosion model by coupling a bathystrophic storm surge model (Freeman et al., 1957; Dean and Dalrymple, 2004) forced by globally-available reanalysis winds (Dee et al., 2011). This model provides water level data based on wind speed, wind direction, coastline angle, and bathymetry. Coastline angle and bathymetry are assumed to remain constant alongshore. The model is quasi-static, and solves reduced equations of motion for storm surge, induced by wind stress and the Coriolis force. The governing equations

are given by

$$g(h+\eta)\frac{\partial \eta}{\partial x} = (h+\eta)fV + \frac{\tau_{\mathrm{sx}}}{\rho} \tag{5}$$

$$\frac{\partial V}{\partial t} = \frac{\tau_{\mathrm{sy}} - \tau_{\mathrm{by}}}{\rho(h+\eta)} \tag{6}$$

where $x$ and $y$ are directed onshore and alongshore, respectively, $g$ is gravitational acceleration [m/s$^2$], $h$ is mean water depth

[m], $\eta$ is the deviation from mean water depth [m], $f$ is the Coriolis frequency [1/s], $\tau_{\mathrm{s}}$ and $\tau_{\mathrm{b}}$ are surface wind and bottom stresses respectively [kg/(m s$^2$)], and $\rho$ is density of seawater [kg/m$^3$]. In the first equation, hydrostatic forces from the storm surge (also referred to in this study as relative water level) in the x-direction (onshore) are balanced by flow V in the y direction (alongshore), and also the wind shear stress component in the onshore direction. In the second equation, the inertial force in the alongshore direction are balanced with alongshore wind surface and ocean bottom shear stresses, which are found using a

drag law (Dean and Dalrymple, 2004). Quasi-static conditions are assumed, such that $\frac{\partial V}{\partial t} = 0$ and onshore flow U is neglected (set to zero). The above equations are solved using a finite difference scheme, and essentially produce a time history of relative water level elevation as a function of changing wind stress.

## 2.3 Model forcing

The forcing for the storm surge model and erosion model come from the ERA-Interim reanalysis dataset (Dee et al., 2011).

Specifically, the 10 m east and west wind speed vectors are used to force the storm surge model, and the sea surface temperature, peak wave period, and significant wave height are used. Winds and sea surface temperature have a 3-hourly temporal resolution. Wave period and significant wave height have a 12-hourly and 6-hourly resolution, respectively. All of these variables were interpolated into hourly timesteps. Changes in wave height, wave period, and sea surface temperature are accounted for when convective heat transfer between the ocean and cliff/beach is calculated by the erosion model (Kobayashi et al., 1999).

When the winds force the storm surge model, it provides water levels on the beach and at the cliff toe for the coupled erosion model. The vector averages of wind speeds and direction over the open water season were also calculated to help analyze the output of the model. The ERA-Interim variables were extracted from the grid cell nearest to each study site (Figure 2). Since



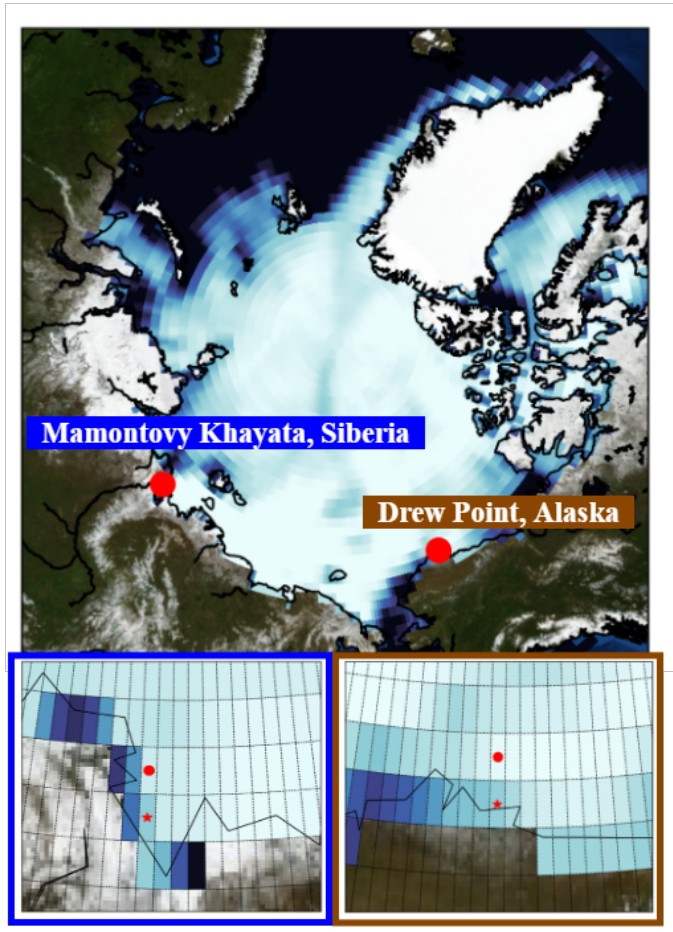

**Figure 2.** Locations of study sites, Mamontovy Khayata, Siberia (blue, left) and Drew Point, Alaska, USA (brown, right). Each are shown in a zoomed-in representation of the ERA-Interim grid, where the red dot indicates the offshore grid cell where sea ice concentration data, wind speed and direction, sea surface temperature, wave height and wave period are extracted for use in the model forcing. The red star below the red dots represent where the actual coastline is in relation to the ERA-Interim grid. Mamontovy Khayata, being located on the narrow Bykovsky Peninsula in Siberia, is not large enough to be represented on the ERA-Interim land mask.

most Arctic erosion occurs during the summer when the coasts are exposed to thermal abrasion by wave action, we use only forcing data over the open water season. To mask the forcing over the ice-covered period, we extracted sea ice concentration 170 from the same grid cells offshore the study sites (Figure 2). When the sea ice concentration had a value of 15% or more, the winds, wave, and sea surface temperature information were masked.



## 2.4 Validation data

Observations of water level were used to validate the storm surge model output. The observed water levels at the MK study site were collected in the summer of 2007-2008 every 15 minutes by a water level gauge (Scheller, 2012). The water levels
were averaged to a 3 hour mean, and the mean of the total time series was subtracted from each timestep value, so that the variability oscillated around 0 m (representative of mean sea level). Monthly tide gauge values are available at nearby Tiksi, but the monthly temporal resolution is not frequent enough to provide meaningful validation of water level values or force ArcticBeach v1.0. However, tide gauge data of a higher frequency (hourly) is available at Prudhoe Bay, Alaska, which is near the other case study site of DP, Alaska (NOAA). The raw tide gauge data is recorded roughly every 6 minutes and was
downloaded as hourly averages. The tide gauge data were further averaged to a 3 hourly mean to correspond with the 3-hourly mean ERA-Interim wind forcing, and then compared to the modelled water level data. To validate the retreat rates, observations of bluff erosion at DP were used (Jones et al., 2018), as well as observed retreat rates at MK on Bykovsky Peninsula (Grigoriev, 2019).

## 2.5 Model calibration

The beach profile along even short stretches of coastlines are highly variable (Overduin et al., 2014), and changes in beach profile directly influence how much water reaches the backing cliff face. Cliff retreat is not activated in the model unless the water level reaches the cliff. Therefore, retreat rates are highly dependent on the water levels reaching the cliff. We have calculated a so-called 'water level offset' that is required for the coupled erosion-storm surge model to reproduce observed erosion rates at each site. This offset is required for two main reasons. The first is that the absolute water depth at the cliff
toe is not known at the study sites (Figure 3), only the water depth relative to local tide gauge datums (where tide gauges are available) are known. The storm surge model calculates water levels relative to still water (no winds) only, which is a reference point that does not exist in reality. The second reason we calculate a water level offset is that it acts as a bulk correction parameter since the model so far only includes primary drivers of Arctic coastal erosion, while secondary physical processes remain to be added, such as thaw slumping and sub-aerial erosion (Overduin et al., 2014). Aside from compensating for the
unknown absolute water level depth, the water level offset can be interpreted as a proxy for the unresolved physical processes driving erosion of Arctic shorelines.

The water level offset was calibrated from annual observed retreat rates for each study site, using a non-linear numerical solver (SciPy.org) with an initial guess of 0.2 m. The offset values were calculated for each year, and the median of the offset from the yearly time series was saved. This median offset value for each site was added to the water levels calculated by the
storm surge model. This sum (water level offset plus modelled water level variability) was then used as the time series of water level forcing for the erosion model.



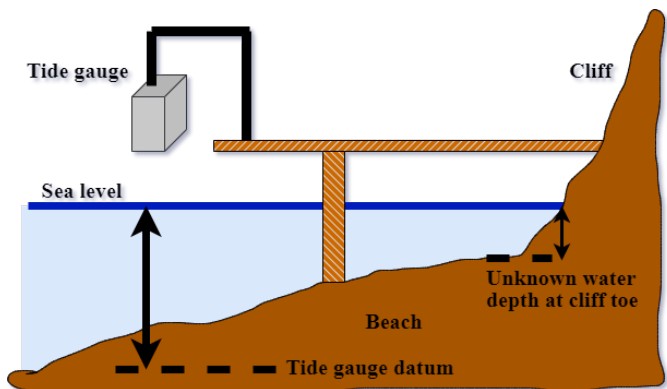

**Figure 3.** Schematic of a reference level for a tide gauge while indicating the water level depth at cliff toe remains unknown due to unknown bathymetry on scales of less than 0.5 m. In this approach, extremely detailed bathymetry information, as well as tide gauges along the entire Arctic coastline, would be required to know the water depths at the cliff toe, which is not feasible. To calibrate ArcticBeach v1.0, our water level offset approach using simulated water level values in response to changing wind speed and direction integrates this issue.

## 2.6 Monte Carlo sensitivity tests

In order to test the sensitivity of the modelled retreat rates, a Monte Carlo approach was used with varying beach and cliff parameters. Each parameter was assigned a realistic range of values, and randomly assigned a value that was within a uniform distribution of this range. For each year, 500 simulations were performed, with the randomly assigned parameter kept constant across all years examined, for each study site during its respective simulation. When the sensitivity of the parameter was not being tested, it was assigned its default value, set according to literature. The default values of these parameters and the referenced ranges that were tested are provided for MK and DP in Table 1.

## 3 Results

### 3.1 Modelled and observed retreat

Observed retreat rates vary from 1.3 - 11.0 m/year at MK and 6.7 - 22.6 m/year at DP (Figure 4 a,b, respectively). The retreat rates given in a cumulative form (Figure 4 c,d for MK and DP, respectively) give a good overview of general model performance on longer timescales, and have been calculated for those years annual observed retreat rates are available.

Over the period where annual observations exist, the cumulative retreat rates agree better with the cumulative modelled retreat at DP (within a few meters) than at MK (roughly 40 m). However, good performance (within a few meters) of individual years can be found for both sites. The frequency of when the model overestimated or underestimated observed retreat followed somewhat of a pattern in MK, where it did not at DP. For example, during the early years of the data record at MK (1997 - 2001), the model agrees well with the observations (error less than 1 m), but in the middle period of the time series (2002 -



**Figure 4.** Observed (orange) and simulated (blue) annual bluff retreat rates (a,b) and cumulative retreat rates (c,d). Values for Mamontovy Khayata are given in (a,c) and Drew Point in (b,d).

2008), the model underestimates observations, and in the later years (2009 - 2018) the model overestimates observed retreat.

This causes cumulative simulated retreat time series to resemble an exponential curve, while the observed cumulative retreat has a more linear curve (Figure 4c).





## 3.2 Storm surge model performance compared to tide gauge data

The storm surge model, providing the water levels due to changing wind conditions to our erosion model, reproduces the observed water level variability relatively well at both locations (Figure 5). Unlike the simulated water levels, the reference

baseline for Prudhoe Bay tide gauge data (blue line in Figure 5b) is mean sea level. Mean sea level does not correspond to a water depth with no winds (which is the reference for our simulated water levels) because mean sea level is also influenced by local currents and larger-scale ocean circulation (e.g. the Alaska coastal current (Talley, 2011) and the Beaufort Gyre at DP). Observed water levels at MK (blue line in Figure 5a) were taken from a depth relative to where the water depth sensor was deployed, which was around 11 m from the surface (Scheller (2012) and Section 2.4). To compare the variability between the

simulated and observed water levels at MK, the baseline of the water level sensor has been set equal to the baseline (relative to 0 m) of the simulated water levels. Bathymetries with a very high spatial resolution are not required for water level simulations. This could prove advantageous for use in areas where nearshore bathymetry must be approximated due to insufficient data. In MK, the water level model is able to reproduce the pattern of observed water level, with the exception of very high peaks and very low troughs (Figure 5a, c) The range of the modelled water levels is 1.2 m and the range of observed water levels is 2.7 m,

with a significant correlation of 0.40. In contrast, at Prudhoe Bay, 3 hourly means of available tide gauge data (recorded roughly every 6 minutes, averaged over every hour) from 2007 - 2016 consistently gave less extreme highs and lows compared to the simulated data (Figure 5d). Since the Prudhoe Bay tide gauge provides values relative to mean sea level, and the storm surge model provides water level values relative to still water depth, an offset between the two datasets is expected. For example, in 2007, the simulated water level values were consistently lower than the observed water level values, but the 3-hourly variability

was still well captured. The range of the modelled and observed water levels are similar, at 1.1 and 1.0 m, respectively, with a significant correlation of 0.64 (Figure 5b).

## 3.3 Coastal winds and modelled water levels

The storm surge model is primarily driven by changes in wind stress. In the Northern Hemisphere, when winds are primarily directed toward the left (as observed from the beach) alongshore or directly offshore during the open water season, a relatively

low water level is expected due to the Coriolis force and wind stress working to push water offshore. This effect becomes apparent during the 2007 open water season at the north-facing coastline of DP, when the winds were most frequently directed offshore (Figure 6a, left panel). This offshore wind pushed the water away from the coast, resulting in an average water level negative relative to what it would have been in calm conditions (Figure 5b and left panel of Figure 6b). In the 2009 open water season at DP, offshore winds were less frequent, while more frequent and stronger north-northeasterly winds (Figure 6a,

right panel) allowed some water to accumulate closer to the beach, but still produced a mean negative value (Figure 6b, right panel). Winds coming from northeasterly directions in 2009 is more typical of DP than offshore southeasterly winds that were observed during the open water season in 2007. Both years had roughly the same open water season, but unlike the 'clean' and well-defined open water period of 2009, 2007 had a false break-up in mid July, as well as a false freeze-up near the end of October (black line in Figure 6b, left panel). A false break-up occurs when ice melts out or breaks off the coast, and then forms



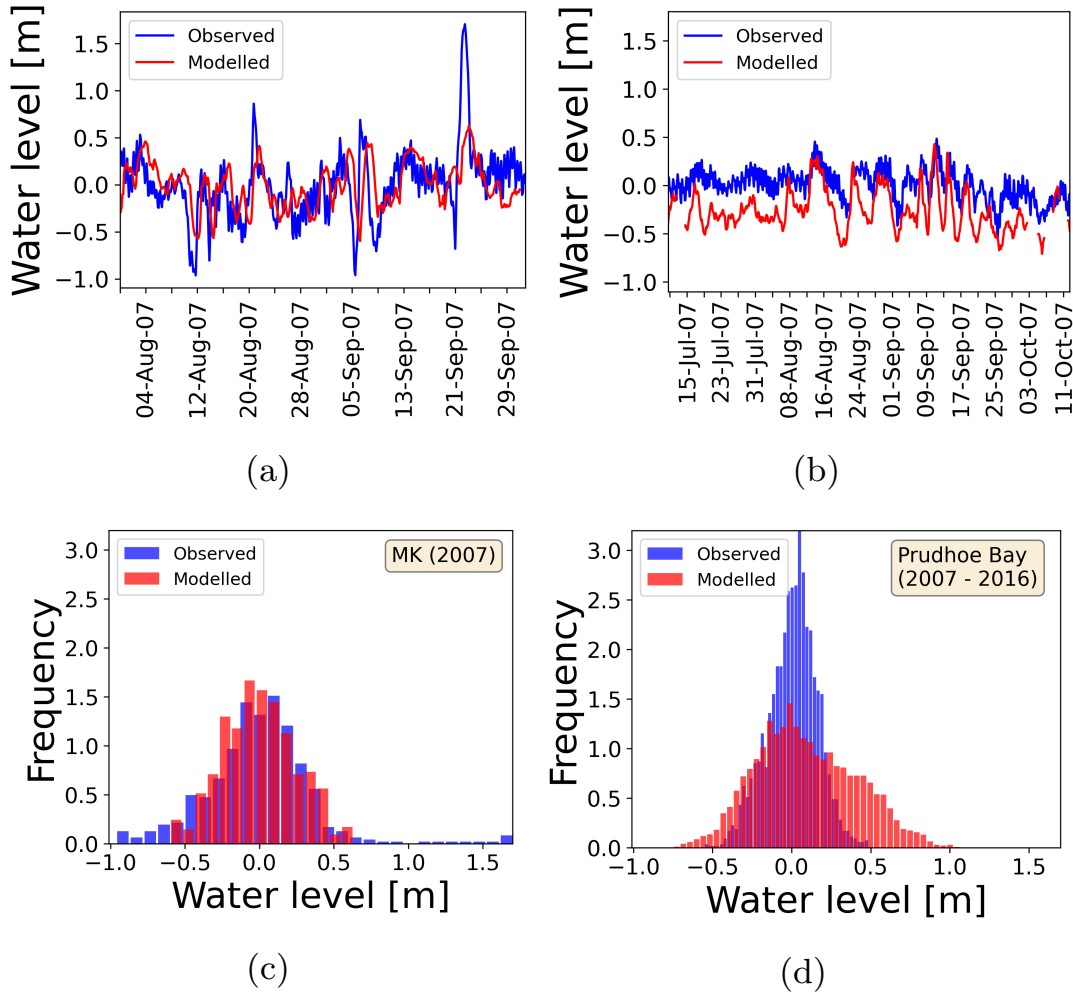

**Figure 5.** Comparison of modelled (red) and measured (blue) water levels. The forcing for the modelled water levels is masked based on sea ice concentration (resulting in a different time period analyzed at each site) and is from the respective offshore ERA-Interim grid cell closest to where the water level validation data was measured near each study site. The modelled water levels have an offset applied such that the mean modelled water level is equal to the observed water level for a,c, and d. In (a) the observed water levels near the MK site are taken from a one-time deployment of a water depth gauge at 71.53°N , 129.56 °E in 2007 (Scheller, 2012). In (b) the observed water levels (blue line) are from the Prudhoe Bay tide gauge (near DP), with data from the year 2007 relative to mean sea level given here as an example, with the corresponding modelled 2007 water levels (red line). (c) shows the frequency of the modelled and observed water levels for MK (comparison only available for 2007) and (d) the full erosion period studied for DP (2007-2016).

or drifts in again before the longer open water season. A false freeze-up is similar, when the ice forms or drifts in at the coast but then returns to open water before the longer ice-covered season (Rolph et al., 2018).



**Figure 6.** Frequency of wind speed and directions with corresponding modelled water levels and sea ice concentrations for selected years at Drew Point. Wind directions are vector-averaged over the open water season. The open water season is defined when the sea ice cover (black line) is below 15% (black dashed line). Wind and sea ice data are taken from ERA-Interim reanalysis.

The MK coastline faces northeast. So, northeasterly winds should generally push water towards shore, raising the water levels near the coast. Onshore winds are more frequent at MK (Figure 7a), compared to winds at north-facing DP (Figure 6a). Consequently, water levels simulated at MK forced under these winds are higher than at DP (compare red mean water level lines in Figures 6b and 7b). The 1999 open water season was roughly twice as long compared to 2002. The open water season was relatively well-defined in 1999 except for 1 false break-up event at the end of June, while 2002 had 14 short false break-up and freeze-up events scattered throughout its short open water season (black lines in Figure 7b).



**Figure 7.** Frequency of wind speed and directions with corresponding modelled water levels and sea ice concentrations for selected years at MK. Wind directions are vector-averaged over the open water season. The open water season is defined when the sea ice cover (solid black line) is below 15% (dashed black line). Wind and sea ice data are taken from ERA-Interim reanalysis.

### 3.4 Variability of water level offsets over a changing open water season

Variability of the open water season during the years with observed retreat rates is higher at MK than at DP (blue bars in Figure 8). At MK, the open water season ranges from 52 - 133 days, and at DP, from 86 - 133 days. Similar to the duration of the open water period, the variability of the derived water level offset is found to be higher at Mamontovy Khataya than at DP (red stars in Figure 8). The range of the water level offset for MK is -0.2 - 2.2 m, and 0.2 - 1.0 m for DP. Due to the more positive skew of water level offsets at MK, the median water level offset (the final calibration value used to force the model)





is further from the mean water levels at MK in comparison to the nearly equal median and mean water level offsets at DP

(compare distances between the black solid and red dashed lines at each of the two sites in Figure 8).

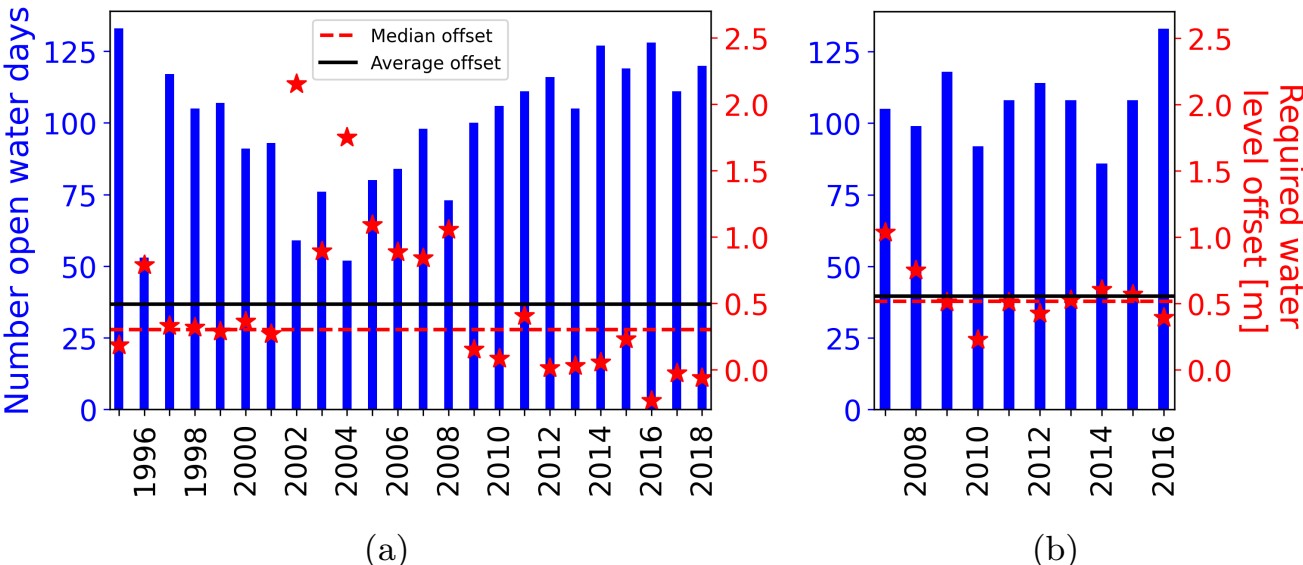

**Figure 8.** The number of open water days (number of days sea ice concentration is below 15%) for (a) Mamontovy Khayata and (b) Drew Point. The average (black line) and median (red line) of the water level offsets for (a) Mamontovy Khayata and (b) Drew Point, required for the model to reproduce observed retreat rates. The yearly required water level offsets are given by the red stars.

### 3.5   Sensitivity to critical model parameters

The sensitivity of the model was analyzed regarding uncertainties for individual parameters including cliff height, cliff angle, and ground ice content. Furthermore, a full uncertainty test was performed within which multiple model parameters (see Table 1) were varied within physically reasonable ranges.

The simulations for DP showed a higher sensitivity of retreat rates to changes in cliff height than the simulations for MK (Figure 9). In general, higher sensitivity of retreat rates to changes in cliff height occurs in the location with the lower initially-prescribed cliff height (DP). At MK, years with higher retreat rates simulated during typical conditions (defined in Section 2.6) show a higher sensitivity of retreat rate to a changing cliff height (1995, 2009-2018 in Figure 9a) than years with lower simulated retreat rates during typical conditions (1996-2008). At both locations, there are noticeably more outliers overestimating

retreat rates than outliers underestimating retreat rates. At DP, the average interquartile range of retreat rate sensitivity to changes in cliff height (Figure 9b) was roughly 10 m and relatively consistent across all years tested, with the exception of 2007 which had a low modelled retreat rate under default parameters. Sensitivity of retreat rate changes in cliff angle is smaller than that of change in cliff height for both study sites (Figure 10). When the simulated retreat rates using default parameters were low (e.g. 1996-2008 at MK, and 2007 at DP, indicated by the blue dots in Figure 10), then the sensitivity to the cliff angle

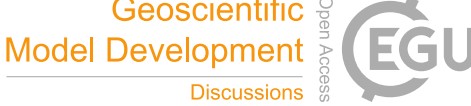

is also low. Sensitivity of retreat rates changes in cliff ice content is similar to that of cliff angle for both sites (Figure 11). While still within the same order of magnitude, the observed retreat rates mostly lie outside of the interquartile range given by all sensitivity tests. This is also true for the full uncertainty runs, where cliff height, cliff angle, unfrozen cliff sediment thickness, coarse sediment volume per unit volume of unfrozen cliff sediment, and cliff ice content were allowed to vary (Figure 12, Table 1).

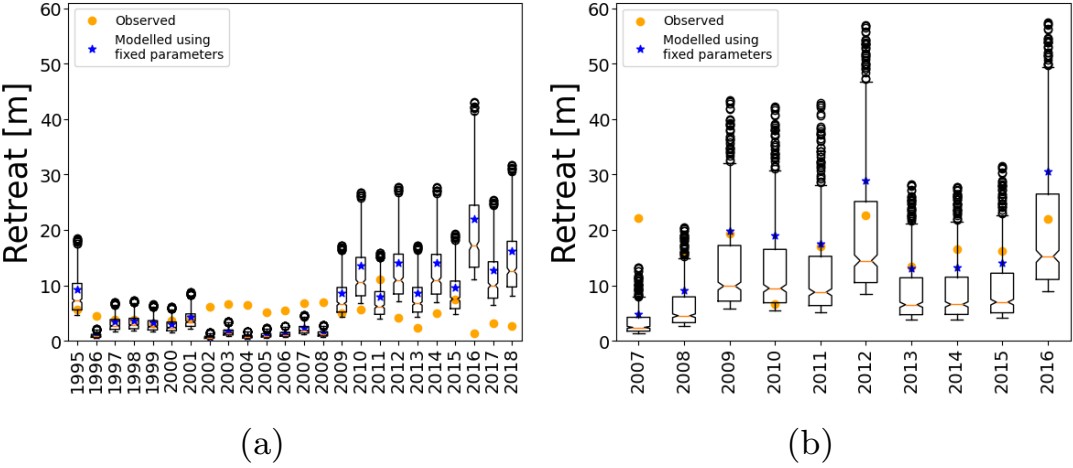

**Figure 9.** Erosion rate sensitivity from changes in cliff height for (a) Mamontovy Khayata and (b) Drew Point. Blue dots indicate the retreat rates simulated under fixed model parameters. Orange dots indicate retreat rates based on observations.

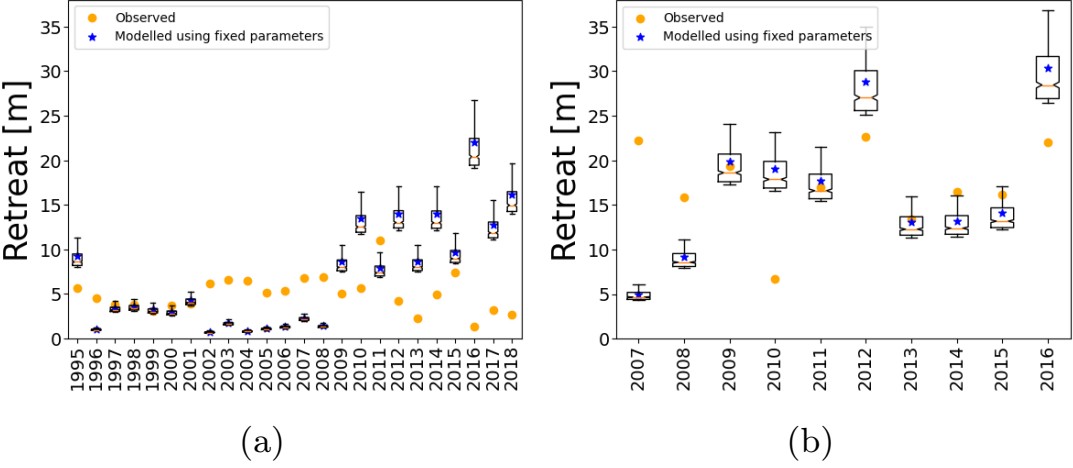

**Figure 10.** Erosion rate sensitivity to changes in cliff angle for (a) Mamontovy Khayata and (b) Drew Point. Blue dots indicate the retreat rates simulated under fixed model parameters. Orange dots indicate retreat rates based on observations.



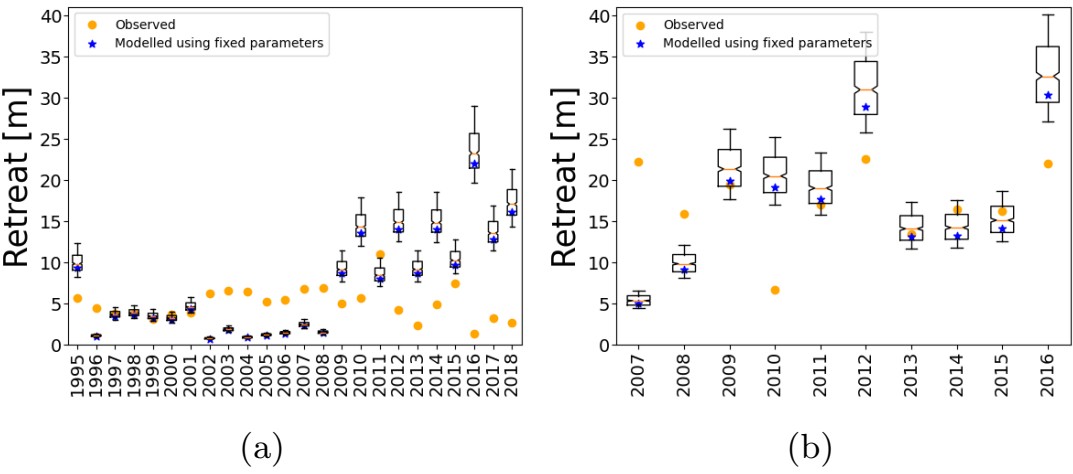

**Figure 11.** Erosion rate sensitivity to changes in cliff ice content for (a) Mamontovy Khayata and (b) Drew Point. Blue dots indicate the retreat rates simulated under fixed model parameters. Orange dots indicate retreat rates based on observations.

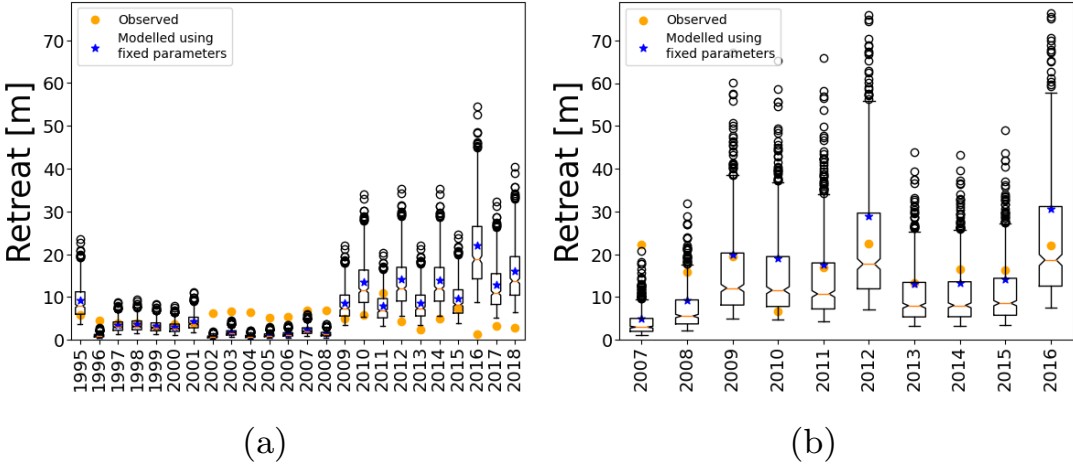

**Figure 12.** Erosion rate sensitivity to changes in cliff height, angle, unfrozen sediment thickness, coarse sediment volume per unit volume of unfrozen sediment, and ice content for (a) Mamontovy Khayata and (b) Drew Point. Blue dots indicate the retreat rates simulated under fixed model parameters. Orange dots indicate retreat rates based on observations.

## 4 Discussion

### 4.1 Effects of calibration on simulated retreat rates

The simulated retreat rates (Figure 4) are highly sensitive to the calculated water level offset forcing (Section 2.5 and red stars in Figure 8). The variability of the simulated water levels and open water season length directly influence model performance





of reproducing observed retreat rates. This agrees well with the results of Barnhart et al. (2014) and Islam et al. (2020) such

that Arctic erosion rates are highly sensitive to ocean water level. An important tuning parameter of our erosion model is the median of the so-called water level offsets that were calculated from a yearly time series (see Section 2.5). A higher skewness of the yearly offset value will naturally result in a median value less representative of the individual yearly points. This is demonstrated, for example, by the median value (red dashed line in Figure 8a) being less representative of individual offsets (red stars in Figure 8a) at MK than at DP, where the median value matches the yearly offsets well (Figure 8b). Essentially,

since this median value (Section 2.5) is then added to the simulated water level variability driven by changes in wind speeds and direction (Section 2.2), how well the median matches the individual, yearly calibrated values will directly reflect model performance during individual years. Indeed, we can see that the retreat rates modelled for DP (the location where the median offset is closer to the mean of the calibrated offset values of individual years) match observed retreat better than at Mamontovy Khayta (Figure 4c,d).

#### 4.1.1 The impact of wind direction on modelled water level and erosion

Unchanging wind vectors result in a constant modelled water level. Given similar open water season lengths, low annual variability in wind speed and direction will result in similar simulated water levels. The water level offset, a tuning parameter used in this model (Section 2.5), is a function of observed retreat rate and wind vectors over a changing open water season. Since the same tuning value (the median of the annually calculated water level offset, per study site) is used across all years, we

can expect ArcticBeach v1.0 to perform better in locations where the median and mean of the annual values used to calculate the tuning parameter are similar. In other words, the skewness of the annual water level offset time series can be a predictor of how well ArcticBeach v1.0 will perform at a given location. At DP, for example, the lower variability of the open water season compared to MK (Figure 8) results is a less positive skew of the water level offset, causing ArcticBeach v1.0 to simulate observed retreat rates better at DP (Figure 4). Causes for low skewness in the annual water level offset time series could be

a more consistent open water season, along with persistent wind speeds and directions, as well as low variability in observed retreat rates. Therefore, ArcticBeach v1.0 will perform best at a coastline that meets these conditions. However, since the tuning parameter is a function of all these different conditions, changes in one aspect can be compensated for by changes in another. For example, given the same observed retreat rate, a similar water level offset would be calculated for a short open water season but strong winds pushing water onshore as a season with a longer open water duration but calmer winds. To describe this idea

in more detail, we now analyze the performance of ArcticBeach v1.0 using the examples of individual years at our two study sites, taking into account the length of the open water season, wind direction, and mean modelled water levels.

ArcticBeach v1.0 simulated the observed retreat rates almost exactly in 2009, while underestimating retreat rates by roughly 23 m in 2007 (Figure 4b). Taking a closer look at the wind directions during these years, the primarily southeasterly winds during the open water season in 2007 (left panel, Figure 6a) push water away from the DP coast more effectively than the

stronger, primarily northeasterly winds of 2009 (right panel, Figure 6a). Given that the duration of the open water season is similar in both 2007 and 2009 (Figure 8b), the differences in wind direction explain why the average modelled water levels in 2007 are lower than in 2009 (Figure 6b). Since the median of the annual time series of the water level offset (Figure 8b) is





closer to the average modelled water level value in 2009 than it is in 2007, this results in a better performance of ArcticBeach v1.0 in 2009 compared to 2007.

At MK, the erosion model underestimates the observed retreat rate of 7 m in 2002 by roughly 6 m, while successfully reproducing the observed retreat of roughly 4 m in 1999 (Figure 4a). In contrast with the similar open water season length at DP in our example years of 2007 and 2009 described above, the length of the open water season at MK for 1999 is slightly less than half of the open water season of 2002. Also, in contrast with our example years at DP, the wind directions in 1999 and 2002 over the open water season are similar at MK in both speed and direction (Figure 7a). This results in a similar modelled

mean water level in 1999 and 2002, and therefore a similar difference to the median water level offset added to the modelled water level variability used to force the erosion model. However, due to the significantly shorter open water season in 2002 (Figure 7b), the cumulative water level reaching the cliff and therefore available to cause erosion during the open water season is much less in 2002 than in 1999. The much shorter open water season understandably leads to a higher required water level offset for the model to reproduce observed retreat, much higher than the median of the offsets over all years (Figure 8a). This

large difference between the modelled average water level and median required water level offset result in an underestimation of observed erosion in 2002 at MK. These examples illustrate how ArcticBeach v1.0 performs under years of variable open water seasons, and suggest that under a more uniform open water season length, ArcticBeach v1.0 will simulate observed retreat closer to reality. With a pack ice cover retreating to the north, including the area of partial sea ice cover (Rolph et al., 2020), we can expect the open water season to become more uniform in duration, and subsequently expect the current setup of

ArcticBeach v1.0 to perform better under projected climate conditions.

## 4.2   The impact of geomorphological cliff and beach parameters on modelled erosion retreat rates

Due to the computationally inexpensive and fast nature of ArcticBeach v1.0, our model can provide a quick and useful tool about which parameters (e.g. cliff height, ice content) are the most important in influencing the rate of cliff retreat. This can be particularly useful to help design experiments for physical wave tank models of partially frozen beach erosion (Korte et al.,

2020). Sensitivity of erosion rates to changes in cliff parameters is high (Figures 9 - 12). Sensitivity of retreat rate to changes in cliff height is also understandably influenced by the ratio of water level change to total cliff height. This is shown by the lower sensitivity to changes in cliff height at the prescribed higher cliffs at MK (Figure 9a), compared with the higher sensitivity of retreat of shorter bluffs found at DP (Figure 9b). Given short bluff heights and high water level forcing, the rate of retreat will tend to increase, as expected by Equation 1 and shown in Figure 9. During years where the median water level offset of the

full time series is higher than the annual offset (e.g. in 1995, 2009, 2010, and 2012-2018 at MK, Figure 8a), the cliff length exposed to seawater (distance of the cliff submerged in seawater from the cliff toe upwards) is overestimated in the final model forcing. Therefore, changes in cliff height ($H$) will result in a greater change in $\frac{\partial R}{\partial t}$ when the cliff length exposed to seawater ($l_c$) is larger (Equation 1). This length is directly proportional to the level of convective heat transfer and thaw of the cliff itself, resulting in retreat (See Section 2.1.1 and Equation 1). Indeed, the highest sensitivities of retreat due to changes in cliff height

occur during those years where the median water level is higher than the annual water level offset at Mamontovy Khayta (1995, 2009, 2010, and 2012-2018, Figure 8a).





Cliff angle is important in our simulations of erosion rates because the angle of the cliff (given the same water depth at the cliff toe) determines the length of the cliff exposed to the relatively warmer seawater, influencing the level of convective heat transfer and subsequent cliff thaw. Similarly, the ice content of the cliff is also directly proportional to how effective the

convective heat transfer applied to the cliff is at thawing the cliff sediment, releasing it onto the beach for subsequent transport offshore (Equation 1). This process is particularly apparent at DP, where changes in cliff ice content are more influential of erosion rates than at MK (Figure 11). This is due to the fact that the erosional seawater covers a greater fraction of the shorter cliffs prescribed at DP than at the higher cliffs MK (Table 1). As found in Hequette and Barnes (1990), cliff and beach parameters alone cannot explain the observed erosion rates, which agrees with our sensitivity test results in this study. Sea ice

gouging, for example, can play an important role in nearshore erosion and accretion (Hequette and Barnes, 1990).

### 4.2.1 Water level offsets as a proxy for unresolved processes

The variability and magnitude of the water level offset (Figure 8) is also a proxy for how much the processes that are not included in the model (e.g. sub-aerial erosion and thaw slumping) play a role in determining the observed retreat rate. A thermal heat flux model, such as CryoGrid (Westermann et al., 2016), can be used to identify the changing thaw depth of the bluff which

is currently a constant in the model. Further investigation is required to derive either an empirical or physical estimate of thaw slumping rates as a function of changes in thaw depth. However, calibration from existing slumping observations (Lantuit and Pollard, 2008) in conjunction with CryoGrid output over the same time periods could lead to such a result. This empirical or physical function would then be incorporated into the rest of the physical processes represented within ArcticBeach v1.0 to give a more complete overview of erosional processes at play at partially-frozen coasts.

### 4.3 From proof-of-concept to pan-Arctic application

There are two routes we can take in the move from applying ArcticBeach v1.0 at the two proof-of-concept study sites as was presented here, to using this model on a pan-Arctic level. The first approach would be to calibrate the water level offset on the rest of the Arctic coastlines, and run the model the same way it was implemented in this study. The second approach would be to calculate the absolute water level depth at the base of the cliff instead of calibrating a water level offset. Assuming

that cliff and beach parameters listed in Table 1 remain constant, future permafrost coastline retreat can be projected with projected forcing data (wind speed and direction, sea temperature, and sea ice coverage) available through global climate models. Nutrient and carbon contents in sediments along the Arctic shoreline are available from databases, so that historical and projected coastline retreat rates can be used to calculate biogeochemical fluxes from land to sea due to erosion (Tanski et al., 2016; Dunton et al., 2006). Using the order of magnitude of erosion rates (Figure 4) provided by ArcticBeach v1.0, in

combination with information about how much nutrients are contained in the eroding material (Tanski et al., 2016), changes in nearshore biogeochemistry could theoretically be estimated. ArcticBeach v1.0 can supply sediment masses deposited in the nearshore zone in an automated fashion to a coupled to a nearshore biogeochemical model, or a biogeochemical module within a greater earth system model such as HAMOCC (Ilyina et al., 2013).



The next step demands the exploitation of pan-Arctic datasets such as Lantuit et al. (2012) which might be used as baseline
tuning data as described in Section 2.5. This potential path that remains to be explored in-depth in future work is to apply
the same methods presented in this study to the rest of the Arctic coastline. Even if we have very coarse temporal resolution
retreat rate data, if covered over a long enough time period (for example, a decade or more) it would theoretically be sufficient
to calibrate the median water level offset (Section 2.5). Such datasets of observed retreat rates are available in Lantuit et al.
(2012) as well as a geomorphological classification scheme for 101,447 km of the Arctic coastline. Using this classification
scheme, we could potentially assign the input parameters of ArcticBeach v1.0 (e.g. cliff heights, ice contents, Section 2.1.3).
These initialization parameters, as well as the varying forcing data along the coastline, could then be used to calibrate the
model and calculate retreat rates for the entire coastline. However, whether or not the model will reproduce a climatology of
observed retreat rates remains to be tested, which would provide further insight on the feasibility of using projected forcing
data to assess pan-Arctic erosion rates under climate warming scenarios.

A second approach to apply ArcticBeach v1.0 on a pan-Arctic level is to eliminate the need to calibrate the modelled water
levels to observed retreat rates. A main reason we must calibrate our model is that we do not know the absolute water depth
at the eroding cliff toe. Anywhere along the Arctic coastline, we are able to calculate a time history of the changes in water
level attributed changing wind speeds and directions. However, these calculated changes in water level are relative to the purely
theoretical baseline of water without winds, and remain to be superimposed on local absolute water levels. Promising results
show that nearshore bathymetry of 10 m can be achieved using satellite data (Caballero and Stumpf, 2019). There is potential to
use geo-referenced water level measurements (SONEL) in combination with methods that provide very high resolution Arctic
coastline bathymetry data (Caballero and Stumpf, 2019) such that calibrating the water levels to observed retreat rates could
be avoided.

## 5 Conclusions

We demonstrate that coupling a reduced order storm surge model to a one dimensional permafrost coastal erosion model pro-
duces realistic coastline erosion rates for two very different locations along the Arctic coastline. The model is solely forced with
globally-available climate reanalysis data, but any type of wind forcing can be used (e.g. coupled to a stand-alone atmospheric
model, meteorological station data, etc). Our final retreat rates are within the same order of magnitude as the observed retreat
rates for both proof-of-concept study sites. In this sense, the model represents the processes dominating permafrost coastline
erosion well. More complex processes controlling spatial and temporal variability in coastline erosion such as thaw slumping
and sub-aerial erosion are not yet implemented but can be added. Although calibrating this model requires knowledge of past
retreat rates, this calibration data can be of a low temporal resolution and already exists in published literature at the pan-Arctic
scale. The requirement for water level calibration can be removed in future work. Since ArcticBeach v1.0 is computationally
inexpensive, it can be used for quick sensitivity studies to evaluate which physical processes and morphological properties of
the cliff and beach are most important in simulating retreat rates of a partially frozen coastline. The simulations performed here
demonstrate that water level on the cliff face is one of the most important aspects driving bluff retreat, supporting the findings

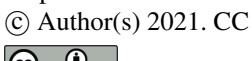



of other studies. Further application to forecast erosion rates using the physical principles applied here is possible through use of projected climate data.

*Code availability.* The model and scripts for analysis can be found at https://doi.org/10.5281/zenodo.4486817

*Author contributions.* RR, ML, and PO designed the study. RR developed the model code and performed the simulations. RR wrote the manuscript, with contributions from all co-authors.

*Competing interests.* The authors declare no competing interests.

*Acknowledgements.* This work was financially made possible by Geo.X, the Research Network for Geosciences in Berlin and Potsdam. Grant number SO_087_GeoX. Further, the work was supported by the Federal Ministry of Education and Research (BMBF) of Germany
through a grant to Moritz Langer (no. 01LN1709A). TR was supported by the National Science Foundation (NSF) award 1745508. HL was supported by the European Union's Horizon 2020 Research and Innovation Programme (Nunataryuk, Grant 773421).



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
