# Peer review of "ArcticBeach v1.0: A physics-based parameterization of pan-Arctic coastline erosion"

_Geoscientific Model Development, 2021_

## Referee Comment (RC1)

**Summary**
Thank you for the opportunity to conduct a review of this manuscript. Here, the authors couple an existing storm surge and erosion model to estimate annual rates of coastal erosion for two study areas in the Arctic (Drew Point, AK and Manmontovy Khayata, Siberia). The authors conclude that they can predict multi-annual cumulative erosion on the same order of magnitude of what has been observed at these sites and that their methodology is an important first-step toward an approach for estimating erosion for pan-Arctic scales.

**Recommendation**
I commend the authors on their writing styles, as evidenced by the small number typographical errors throughout the manuscript. However, this work hosts a multitude of technical issues, most notably the study's methodology and conclusion based therefrom. My feeling is that it does not warrant publication. For this reason, I have limited my review to two major comments, as opposed to more detailed in-line comments.

**Major Comments**
The authors highlight that "the most important root causes of Arctic shoreline change can only be gained through careful evaluation of the physical processes involved" and yet make no such effort for their own study. For example, one of the two sites where the authors apply their model is Drew Point, AK. Here, it is well known that permafrost blocks bound by ice wedges topple onto the beach due to an undercutting process that is facilitated by storm surge (i.e., "thermo-abrasion"). This reality is in stark contrast with the incremental style of bluff retreat associated with the model of coastal erosion employed by the authors (Figure 1). I understand that the authors ultimately wish to exercise their modeling framework elsewhere, but what is the scientific value of applying such a model to a place like Drew Point? My feeling is that Drew Point is not an appropriate location to apply or test the erosion model the authors use in this study.

My biggest concern regarding the validity of this study is the lack of an error analysis of the model outputs (i.e., annual rates of erosion). The model predictions are higher and lower than the observations and in a somewhat chaotic fashion (Figure 4a-b). In many cases, the model predictions are several factors (approaching an order of magnitude) off. Given that the calibration of the model includes an input of historical retreat rates, is this level of error acceptable? What explains the seemingly non-systematic trends in model error?

**I calculated a negative value for the Nash-Sutcliffe Model Efficiency (EF) statistic using the measured vs. modeled erosion rates reported for Drew Point in this study, which indicates that the mean of the Drew Point observations is a better predictor of annual erosion than the author's model. This back of the envelope calculation with a widely used error analysis metric underscores a potentially major issue regarding the predictive power of the author's model.**

The EF is given by:

$$EF = \left[ \sum_{i=1}^{n} \left( O_i - \overline{O} \right)^2 - \sum_{i=1}^{n} (P_i - O_i)^2 \right] \Big/ \sum_{i=1}^{n} \left( O_i - \overline{O} \right)^2,$$

where $P_i$ are the predicted values, $O_i$ are the observed values, n is the number of samples, and $\overline{O}$ is the mean of the observed data. The EF statistic ranges from 1.0 to $-\infty$, with 1.0 indicating a perfect match between $P_i$ and $O_i$ and EF less than zero indicating that $\overline{O}$ is a better model than $P_i$ for simulating $O_i$.

Without a formal error analysis or comparison to another erosion model, it difficult to argue that this study has advanced our understanding of Arctic coastal erosion processes or produced meaningful insights for the communities that are vulnerable to this environmental problem.

---

## Referee Comment (RC2)

[referee-annotated manuscript omitted]

---

## Referee Comment (RC3)

**ArcticBeach v1.0: A physics-based parameterization of pan-Arctic coastline erosion**

Rebecca Rolph[1,2], Pier Paul Overduin[1], Thomas Ravens[3], Hugues Lantuit[1,4], and Moritz Langer[1,2]

[1]Alfred Wegener Institute Helmholtz Centre for Polar and Marine Research, Telegrafenberg A45, 14473, Potsdam, Germany
[2]Geography Department, Humboldt-Universität zu Berlin, Unter den Linden 6, 10099, Berlin, Germany
[3]University of Alaska Anchorage, 3211 Providence Dr., Anchorage, AK 99508, United States
[4]University of Potsdam, Am Neuen Palais 10, 14469, Potsdam, Germany

**Correspondence:** Rebecca J. Rolph (rebecca.rolph@awi.de)

**Abstract.** In the Arctic, air temperatures are warming and sea ice is declining, resulting in larger waves and a longer open water season, all of which intensify the thaw and erosion of ice-rich coasts. This change in climate has been shown to increase the rate of Arctic coastal erosion, causing problems for industrial, military, and civil infrastructure as well as changes in nearshore biogeochemistry. Numerical models that reproduce historical and project future Arctic erosion rates are necessary to understand

5  how further climate change will affect these problems, and no such model yet exists to simulate the physics of erosion on a pan-Arctic scale. We have coupled a bathystrophic storm surge model to a simplified physical erosion model of a partially frozen cliff and beach. This Arctic erosion model, called ArcticBeach v1.0, is a first step toward a parameterization of Arctic shoreline erosion for larger-scale models, which are not able to resolve the fine spatial scale (up to about 40 m) needed to capture shoreline erosion rates from years to decades. It is forced by wind speeds and directions, wave period and height, sea

10  surface temperature, all of which are masked during times of sea ice cover near the coastline. Model tuning requires observed historical retreat rates (at least one value), as well as rough nearshore bathymetry. These parameters are already available on a pan-Arctic scale. The model is validated at two study sites at Drew Point (DP), Alaska, and Mamontovy Khayata (MK), Siberia, which are respectively located in the Beaufort and Laptev Seas, on different sides of the Arctic Ocean. Simulated cumulative retreat rates for DP and MK respectively (169 and 170 m) over the time periods studied at each site (2007 - 2016, and 1995 -

15  2018) are found to be within the same order of magnitude as observed cumulative retreat rates (172 and 120 m). Given the large differences in geomorphology and weather systems between the two study sites, this study provides a proof-of-concept that ArcticBeach v1.0 can be applied on very different partially frozen coastlines. ArcticBeach v1.0 provides a promising starting point to project the retreat of Arctic shorelines, or to evaluate historical retreat in places that have had few observations. Further, this model can provide estimates of the flux of sediment from land to sea for Arctic nearshore biogeochemical studies, while

20  leaving an opportunity for further development of modelling the physics of a partially frozen shoreline.

**1 Introduction**

Arctic coastlines are increasingly vulnerable to erosion due to warmer temperatures (Biskaborn et al., 2019) through the destabilization of frozen cliffs, reduced sea ice protection from bigger waves (Casas-Prat and Wang, 2020; Overeem et al.,

**Commented [AS1]:** Comm4
Yes, coastal erosion is one of the main natural hazards when it comes to infrastructure in the Arctic. Yet, it is rear when we place the infrastructure at ice-rich coasts, as they are known to have highest erosion rates. We normally try to place infrastructure at more stable morphological types, such as barrier islands, river deltas, etc.

If so, then motivation for the article could be somewhat shifted towards the biogeochemistry.

**Commented [AS2]:** Comm1
Yes, frozen cliff and beach a partially frozen during summer and may be fully frozen during winter. These are normal variations in the state of a permafrost coast. The model deals with particular conditions of a permafrost coast. Hence, I would suggest first to outline that the model is handling a dynamics of a permafrost coast, and then, if the authors think that that is necessary, to point out that the model focused on partially frozen cliff and beach.

**Commented [AS3]:** Comm2
Arctic coastlines include different morphologies (+ sandy beaches, rocky coasts, river deltas), not only ice-rich coasts. Is the ice/rich coasts the dominant morphology in the Arctic? Perhaps such coasts constitute some 40-50 %, then one could say that the model will support such large-scale models when it comes to unlithified coasts.

**Commented [AS4]:** Comm3
Is sea surface temperature masked during the times of ice cover? It think that it does not as it is a boundary condition defining permafrost temperature in the nearshore and the shore/shoreface.

**Commented [AS5]:** Please Comm2 One may suggest to still outline the morphologies where the model could be applicable.

**Commented [AS6]:** Please see Coom1

**Commented [AS7]:** See Coom2

**Commented [AS8]:** See Comm2. I again would like to mention that there are also other morphologies in the Arctic. More sever wave climate will for sure lead to stronger erosion on sandy beaches. And sandy beaches, at least in some cases do not have frozen cliff in summer due to deep active layer.

[Figure]

2011), especially as freeze-up becomes delayed further into the fall storm season. Large-scale atmospheric patterns have

25  been recently attributed to driving the variability of Arctic shoreline erosion (Nielsen et al., 2020) and statistical methods might therefore show promising results to simulate erosion rates. However, understanding the most important root causes of Arctic shoreline change can be only gained through careful evaluation of the physical processes involved. Although extensive process-based models exist (Bull et al., 2020; Ravens et al., 2017, 2012; Hoque and Pollard, 2009; Barnhart et al., 2014) these have only been designed for very specific stretches of coastline and mostly focused on the quickly eroding Drew Point

30  and greater southern Beaufort coastline. These models require extremely detailed initialization data and only pertain to their respective stretch of coastline. These types of models are thus not designed for use on a pan-Arctic level where detailed data on geomorphological characteristics and bathymetry are not available. In addition, notch erosion (undercutting of a steep bluff by water or waves) is a key aspect in their formulation of the coastline retreat process. While this process is important in some locations along the Arctic, notch erosion does not apply on a pan-Arctic scale (Lantuit et al., 2012). Further, most existing

35  erosion models are computationally expensive and require long run times, not suitable for efficient physical modelling on pan-Arctic erosion scale. Therefore, the need remains to form a physics-based numerical model that can be applied across all partially frozen shorelines. We present, for the first time, a general numerical erosion model that can serve as a starting point for a physics-based parameterization of Arctic shoreline erosion in earth system models.

The processes involved in Arctic shoreline erosion are different than their mid- and low- latitude counterparts due to the

40  cold temperatures and presence of ice and frozen soils. Shorelines along the Arctic can be frozen and connected to landfast sea ice (Mahoney, 2018), protecting the bluffs and beaches from abrasive wave action. However, strong winds and storm surges can also push ice roughly onto shore, causing erosion, debris influx, and significant destruction of infrastructure and cultural sites (Bogardus et al., 2020). During the summer, the open water period allows for relatively warmer water to thaw the submerged part of the beach, and warmer air temperatures to thaw the exposed part of the shoreline. Thawing shorelines are

45  especially vulnerable to erosion (Aré, 1988), and climate change accelerates this process due to the lengthening open water season and higher sea and air surface temperatures (Barnhart et al., 2014). Social and economic costs of erosion are high, with entire villages having to relocate (Albert et al., 2018; Hamilton et al., 2016). Nearshore biogeochemistry is also heavily impacted by nutrient-laden sediment supplied into the Arctic Ocean, with roughly one third of the Arctic Ocean primary production supported by riverine and coastal sediment inputs (Terhaar et al., 2021). Further, thawing and eroding coastlines

50  can exacerbate climate change by releasing previously sequestered carbon from the soil into the atmosphere (Vonk et al., 2012; Fritz et al., 2017).

The paper set-up is defined as follows. In Section 2, we describe the erosion model and the physical mechanisms and associated initialization parameters included for simulating the erosion of a partially frozen cliff and beach. Next, we describe the water level model, and how it uses wind forcing to generate a time history of relative water levels at the coastline, which

55  are then used to drive the erosion model. Data used for the validation of both the erosion and storm surge model components are also provided. In Section 3, model results and validation are given, along with model sensitivity to critical parameters. Sections 4 and 5 provide a discussion of the results and conclusions.
* * *
**Commented [AS9]:** Comm6. What about the roles of coastal types? Why it is not mentioned when it comes to the variability of erosion rates?

**Commented [AS10]:** Comm7. It sounds somewhat confusion to mask different geomorphologies by referring to locations. See Comm2

**Commented [AS11]:** Why the authors do not mention typical coastal types, and corresponding coastal processes, which would be thermal abrasion and thermal denudation when it comes to unlithified ice-rich coasts?

**Commented [AS12]:** Comm9. Were those villages placed on an ice-rich coast? Probably some of them were.

**Commented [AS13]:** Comm15. When it comes to the motivation behind this article and the biogeochemistry, would be good to refer to Lantuit et al. (2012) and point out that model can be helpful for moving from a static definition of organic carbon (as Lantuit et al. did) to a dynamic.

The Arctic Coastal Dynamics Database: A New Classification Scheme and Statistics on Arctic Permafrost Coastlines

[Figure]

[Figure]

**2 Methods**

We have coupled the framework of an existing 1-D Arctic coastline erosion model (Kobayashi et al., 1999) with a bathystrophic
60  storm surge model (Freeman et al., 1957), forced by wind speed and direction, and initialized using existing bathymetric in-
formation of our study sites. The idealized set-up of the erosion model (Figure 1) includes a beach and cliff profile, assuming
uniform conditions alongshore. Conceptually, the model simulates thawing of the beach and cliff sediments according to con-
vective heat transfer controlled by water level and temperature. Thawed material is assumed to be prone to erosion depending
on water level and wave action. The process of mass transfer is simulated by emulating a cascade of cliff erosion, beach depo-
65  sition, and beach erosion. According to the resulting mass balance, the beach and cliff profiles are adjusted assuming constant
beach and cliff inclination. Small scale processes such as niche formation are neglected in this coarse-scale approach. Further
description of the beach and cliff model parameters are given in Section 2.1.

[Figure]

**Figure 1.** Model sketch illustrating basic physical model parameters (black) and processes (red). Wind forcing, masked during times of sea
ice cover, is taken from the ERA-Interim reanalysis (Dee et al., 2011) dataset to force a coupled storm surge model. This provides water
level data to the erosion model, driving the bluff retreat and beach erosion through a heat and volume balance. Sea surface temperature, wave
height, and wave period are also taken into account, as well as the prescribed cliff and beach parameters of volumetric ice content, sediment
grain size, cliff height, thaw depth, and cliff and beach angle.

**2.1 Erosion model**

The erosion model used in this study is constructed from heat and sediment volume balances in order to predict horizontal cliff
70 retreat and vertical erosion of a fronting beach. A full description of the framework for this model can be found in Kobayashi
et al. (1999), but we provide a brief overview of the main driving mechanisms here. Wave action and water levels drive
convective heat transfer, and thaw ice-bonded sediments comprising the cliff and beach. When cliff sediment, with its initially
prescribed coarse sediment fraction, is released via melting ice between grains of sediment, this coarse sediment is deposited
onto the beach, while the remaining fraction of cliff sediment (the fine sediment) is assumed to be transported offshore by
75 the seawater. The amount of coarse sediment (defined by a grain size threshold) that remains on the beach is determined by
a volume balance. The volume balance is defined as follows: the rate of coarse sediment transport transported away from
the beach cannot exceed a so-called potential sediment transport rate that is determined largely by the beach angle and water
level. In general, steeper beach angles and higher water levels lead to higher potential coarse sediment transport rates away
from the beach and offshore. Flat beaches and low water levels will result in a low amount of coarse sediment that could be
80 transported offshore. More detail of modelled mechanisms driving cliff and beach erosion are given in the subsequent sections
(Sections 2.1.1 and 2.1.2) and also in Kobayashi et al. (1999).

> **Commented [AS14]:** Comm. Would be good to mention that conduction is not taken into account.

**2.1.1 Cliff erosion**

The rate of the cliff retreat is determined by the heat transfer into the exposed frozen cliff face assuming isothermal frozen
sediments at freezing temperature (0 C). The rate of cliff retreat ($\frac{\partial R}{\partial t}$) is, thus, defined by the rate of melting of interstitial ice
85 and subsequent release of cliff sediment determined by the energy supplied divided by the energy required to thaw the part of
the cliff face that is exposed to seawater. This expression is given by

> **Commented [AS15]:** Comm12. Sediments in the coastal zone in the Arctic are normally saline, hence freezing temperature is lower that 0 C.

[revised manuscript text omitted]

**Commented [AS20]:** Comm19. 10 m/s ? What defines the selected value of 10 m/s/?

It is known that lower wind speeds are also capable to generate a storm. It is Ok to use 10 m/s, but one should then outline that this is somewhat a characteristic value. Just

p
e
n
w
at
er
se
as
o
n
w
er
e
al
so
ca
lc
ul
at
e
d
to
h
el
p
a
n
al
y
ze
th
e
o
ut
p
ut
of
th
e

model. The ERA-Interim variables were extracted from the grid cell nearest to each study site (Figure 2). Since

[Figure]

[Figure]

**Figure 2.** Locations of study sites, Mamontovy Khayata, Siberia (blue, left) and Drew Point, Alaska, USA (brown, right). Each are shown in a zoomed-in representation of the ERA-Interim grid, where the red dot indicates the offshore grid cell where sea ice concentration data, wind speed and direction, sea surface temperature, wave height and wave period are extracted for use in the model forcing. The red star below the red dots represent where the actual coastline is in relation to the ERA-Interim grid. Mamontovy Khayata, being located on the narrow Bykovsky Peninsula in Siberia, is not large enough to be represented on the ERA-Interim land mask.

most Arctic erosion occurs during the summer when the coasts are exposed to thermal abrasion by wave action, we use only forcing data over the open water season. To mask the forcing over the ice-covered period, we extracted sea ice concentration

170 from the same grid cells offshore the study sites (Figure 2). When the sea ice concentration had a value of 15% or more, the winds, wave, and sea surface temperature information were masked.

Commented [AS21]: Comm20. It would be good to clarify then that, in fact, some of the winter storms might have been taken into account (which brings us closer to the reality).

[revised manuscript text omitted]

---

## Community Comment (CC1)

[supplement omitted: unrelated document]

---

## Author Comment (AC1)

**Author Response to CC1.**

**The comments in CC1 by Jennifer Frederick are in font color black. The authors' responses are in green.**

We thank Jennifer Frederick for going through our model results in detail. We are very much aware of the current model development efforts in which she is involved and value her opinion in improving our model. She has proposed, with her own analysis, that: ''ArcticBeachv1.0 cannot predict the annual erosion rate any better than a random number generator can.'' Seeing this as a rather crippling statement to our work, we have carefully gone through her code and analysis, and respectfully assert that this statement is not supported nor reflected by what she has provided in her analysis.

We assert:

- Contrary to what Jennifer Frederick has claimed, she did not compare our model results with random numbers. Instead, she used an unrealistically-constrained array of values having limits of 10% within observed retreat rates.
- We would like to point out that *any model* or hindcast would be out-performed by a random number generator as long as the random numbers being generated were constrained to a close enough value to the observations.
- Given that observed retreat rates themselves can have a much greater error than 10% and are difficult to assess (Lantuit et al., 2008, Jones et al. (2018)), we find it an unfair and misleading claim to assert that the results of our model are no better than choosing random numbers when the 'random' numbers were chosen closer to observations than the uncertain error bar on the observations themselves. Further, Jennifer Frederick does not provide any reasoning for choosing a 10% threshold of deviations from observed retreat rates as her definition for her 'random numbers'. It is not mentioned in the main body of her comment that her 'random' numbers are constrained by this unrealistic threshold, well within the error of observations. It is instead left to the reader to go through the methodology of the supplemental section in her community comment, or find the line in her model code. This tactic is potentially damaging in the sense that it could lead to misconceptions of our paper if the reader is unwilling to go through the methodology supplement of community comments in a discussion forum.
- Our model includes essential physics that cannot be produced by a random number generator. For example, water levels are essential in driving coastline retreat (Barnhart et al., (2014), Kobayashi et al. (1999) and mentioned in Section 1, line 43-46 of the manuscript). ArcticBeachv1.0 is able to calculate relative water levels using wind speed, wind direction, bathymetry, and coastline angle. In light of declining sea ice cover and lengthened open water season, especially in those locations where freeze-up is being delayed further into the windy fall storm season, the importance of including a physical representation of changes in relative water levels due to wind forcing is paramount. Water levels have long been known to be a driving factor in erosion, especially erosion of partially-frozen coastlines present in the Arctic (Aré (1988), Casas-Prat and Wang (2020)).
- ArcticBeachv1.0 is the coupling of two widely-known and well-cited physics-based numerical models (Kobayashi et al. (1999) and Freeman et al. (1957)). It is the first time a water level model has been coupled to a simplified Arctic erosion model with the aim of developing a computationally efficient physics-based parameterization of arctic erosion.

- o It is the first time such an approach is used that does not focus on one segment of coastline, such as the highly specialized processes occurring at Drew Point, Alaska. This is mentioned in the manuscript in Section 1, lines 27-38.
- o To say this coupled model is 'under-developed' suggests that a state-of-the-art similar approach already exists, and ours is less developed than state-of-the-art.
- o Since it is the first time an arctic erosion model has been developed such that it can simulate retreat on diverse types of coastlines in a computationally-efficient manner, we argue that ArcticBeach v1.0 sets the state-of-the-art in developing a parameterization of arctic shoreline erosion based on physical principles.

Review of ArcticBeach v1.0: A physics-based parameterization of pan-Arctic coastline erosion
by Rebecca Rolph et al.
Reviewed by Jennifer M. Frederick, Sandia National Laboratories,
jennifer.frederick@sandia.gov

Summary
This manuscript describes a model for Arctic coastal erosion that is based on a simplified physical erosion model of a partially frozen cliff and beach, coupled to a storm surge model. It is presented as a first step toward a parameterization of pan-Arctic shoreline erosion at a coarse spatial scale for capturing erosion rates on the order of years to decades. It uses physical data as boundary conditions, such as wind speeds and directions, wave period and height, and sea surface temperature, as well as accounting for sea ice cover. The authors claim the new model provides a promising starting point to project the retreat of Arctic shorelines, or to evaluate historical retreat in places that have had few observations.

General Summary of Comments
I do not recommend publication in its current form. The model presented (ArcticBeachv1.0) is under-developed and the authors have not shown that this model has any predictive skill that outperforms a random number generator (proof described in detail in my review). For transparency, I have also included the Python script which performs this analysis.

We have gone through Jennifer Frederick's random number generator in her script mentioned above (see the line of her code pasted below in green) and it is not stated in the text of her detailed review (including in her description explaining what a null hypothesis is to the authors) why she tuned the 'random' numbers to generate random values within 10% of the observed retreat. The 10% used in her script refer to the error reported by authors in calculating erosion rates based on remote sensing imagery. The error of observations calculated for remote-sensing based computations of retreat are generally based on equations attempting to encapsulate all errors associated with the manipulations of the operator and of the georeferencing process (see Lantuit et al., 2008, 2011; Günther et al., 2013, Jones et al. 2009). The total error or dilution of accuracy is the root of the sum of all squared error factors (RMSE from georeferencing, digitizing error, pixel resolution) divided by the number of observation years. The individual error factors remain the same independently of the period of observation, while the number of years obviously varies. The total error will therefore vary drastically based on the observation period. In other words, the error reported in these studies relates to longer periods of observations and not to individual years. Previous studies have reported on errors varying between 4 to greater than 25% (see Lantuit et al., 2008, 2011;

Günther et al., 2013, Jones et al. 2008, 2009)., but these errors could be much greater for shorter periods of observations. At Drew Point the total error varied between 4 and 9% (Jones et al. 2009) but would be greater if the observation periods were shortened. This shows that the 10% error chosen here does not reflect the error actually observed for individual years and actually does not relate to one specific period of observation. In fact, Jones et al. (2018) indicate that "it is difficult to accurately assess errors in erosion rate measurements associated with this type of analysis." Finally, the error for observations at Drew Point should be considered as among the lowest along the whole arctic coastline, and is an inaccurate depiction of observational error for average arctic coastal erosion. The selection of a 10% value for error is therefore highly subjective and not systematic.

We would like to re-iterate any model would be deemed out-performed by a random number generator as long as the random number generator was constrained to a close enough value to the observations. While ArcticBeachv1.0 is calibrated with observed values, it is not a random number generator. It is based on a previously published, and well-cited, physical model of Arctic shoreline erosion (Kobayashi et al., 1999). Thus, unlike a random number generator, it provides the opportunity to add more physical processes not yet captured in this study. Additionally, as mentioned in Section 4.3 lines 405-413, the possibility to avoid water level calibration in the future could be implemented in future work.

The line in the code provided by Jennifer Frederick, which constrains the random number generator within 10% of the observed retreat rates, is copy-pasted below:

```
randomlist_DP.append(random.uniform(min(obs_DP)*0.90,max(obs_DP)*1.10))
```

As with all numerical physics-based models, the opportunity exists to add more physical processes with the goal of reproducing model output that more closely resembles reality. One of the primary uses of numerical models is to develop our understanding of the real world, and to predict what might happen in the future. A random number generator that is constrained to past observations is not used in place of numerical models because we do not have observations of the future. While data assimilation is a great tool, it is not possible to solely rely on observations to predict future processes --- we need to understand the physics involved. Or, indeed, if one would like to look at the sensitivity of a certain process, we would have to use a model with multiple processes in order to (for example) effectively 'turn off' that one process we are interested in and see what impact that has on the results. Such an understanding is not possible by using only a random number chosen well within the error of observations. Sensitivity studies to different coastline properties has been conducted and those results explained in Section 3.5 'Sensitivity to critical model parameters'.

One such process that is important for erosion is water level, and we are able to calculate the relative water levels at a very low computational cost, at any point along the Arctic coastline, from globally-available reanalysis wind speed and direction, or, for example, CMIP projected winds. This is done with a well-known and well-cited storm surge modelling approach, explained in Section 2.2, lines 139-157. We argue that a random number generator is in no position to calculate physically-relevant variables that drive Arctic coastal erosion, and our model provides establishes the state-of-the-art of computationally-efficient coupled storm-surge Arctic erosion processes.

My suggestion to the authors is further development of the model and resubmission for publication at a later date and after further collaboration and consultation with peers in this research field.

We view the assumption above that we have not collaborated with peers in this research field as presumptuous. Further, this comment does not refer to any part of the scientific scope of the manuscript.

One benefit of the model presented is its low computational cost. If the low computational cost can be maintained while improving its ability to robustly predict coastal retreat rates, this would represent a ground-breaking advance in the field!

We appreciate that Jennifer Frederick sees our approach as one that is worth developing further, and would like to add that ArcticBeach v1.0 represents a way forward not only in the field of arctic erosion, but also those fields involved in nearshore carbon cycling and biogeochemistry (referenced in Section 4.3, lines 387-393 of the manuscript).

The results summarized in Figure 4 show the modeled annual and cumulative retreat at Mamontovy Khayata (MK) and Drew Point (DP) vs observations at each site. At first glance, the modeled retreat looks poor, but an error analysis was not provided to quantify model performance. For any predictive model, a thorough analysis of model predictive skill is required to evaluate its performance and ability to make reliable, robust predictions. One of the simplest routines is to test model predictions against a random prediction. If the model has good predictive skill, it should outperform a prediction generated at random within a plausible range of possible outcomes. This is essentially like posing the null hypothesis and showing that the model can disprove the null hypothesis. In this case, the null hypothesis states that, 'ArcticBeachv1.0 cannot predict the annual erosion rate any better than a random number generator can.' This null hypothesis would be disproved for any possible model as long as the random number generator is constrained to a close enough range to observations. If all models could be constrained so closely by observations at the same time as knowing what values will be observed in the future, any numerical model hindcast or prediction is irrelevant. Given that Jennifer Frederick unrealistically constrained her random number generator closer to observed values than error of the observations themselves, we do not find her analysis relevant in a scientific context. If the ArcticBeachv1.0 model can predict annual erosion rate statistically significantly better than a random number generator, then it can rightfully claim predictive skill. My concern here for both locations is that, while there are a few years where modeled erosion matched observed erosion fairly well, there are also many years in this time series where the erosion is far outside of the running average. In these years, a model with high predictive skill should be able to reproduce the trend, if it has captured the correct physics. However, the ArcticBeachv1.0 model predictions end up under- or over-estimating the retreat, in the OPPOSITE direction just as many times as they estimate the retreat in the CORRECT direction (above or below the mean retreat).

The conclusion from the analysis for predictive skill (described in full detail below) shows that the ArcticBeachv1.0 model has no predictive skill at the DP location, and has inverse predictive skill at the MK location. Based on the error analysis, I disagree with the authors, as stated in the abstract, that the ArcticBeachv1.0 model provides a promising starting point to project the retreat of Arctic shorelines, or to evaluate historical retreat in places that have had few observations. The results of this analysis at both locations indicate that the model in its current form is under-developed, and cannot be relied upon to provide robust and skillful

predictions for coastal retreat rates in the Arctic more than a randomly generated number can (in the case of the DP location) nor can be relied to provide a prediction in the correct trend direction (in the case of the MK location). These statements are misleading and do not provide the full scope of the reviewer's case for 'random number generator'. The argument that has been made multiple times by the reviewer that the ArcticBeach v1.0 is no more useful at evaluating erosion than choosing random numbers could be applied to any model, not just ArcticBeach v1.0.

We would like to re-iterate to the point that if all models could be constrained so closely by reality (or, as done in Jennifer Frederick's analysis, even closer to reality than the error of what we are able to observe) any numerical model hindcast is irrelevant.

Detailed Analysis

I performed an analysis on the modeled retreat vs the observed retreat to quantify the error. I used the mean squared error (MSE) of the annual retreat predictions as the performance metric. The MSE will penalize large differences between predicted and observed values more so than small differences, which is appropriate in this case because swings in retreat far outside the normal or average erosion behavior signify major disruptions in erosion drivers, which is what we want to capture with a robust, skillful predictive model. It is also thought that future conditions will become more extreme as climate changes in the Arctic, and thus erosion may continue to behave erratically. The mean squared error is defined as

$$MSE = \frac{1}{N} \sum_{j=1}^{N} (M_j - O_j)^2$$

where N is the number of retreat predictions with available retreat observations, M is a modeled retreat prediction, and O is a corresponding retreat observation for the model prediction. For this test, the MSEs for each location for ArcticBeachv1.0 vs observations are shown in Table 1. For the MK location, the MSE of the annual retreat between 1995 - 2018 was 125.48 m2, and for the Drew Point location, the MSE of the annual retreat between 2007 - 2016 was 61.55 m2. A perfect prediction for every year would yield an MSE of 0 m2 at both locations.

[Table 1 is given in original CC1].

Next, the model predictions are tested against a random number generator to judge predictive skill and give meaning to the MSE values calculated for ArcticBeachv1.0. For the MK location, a random number from within a plausible range of retreat was generated for each year using Python's random package. The range in retreat was calculated as the minimum and maximum of the observed retreat data with a 10% envelope (e.g. 1.18 m – 12.04 m). The MSE for the randomly chosen annual retreat was calculated against the observations. This numerical experiment was performed 5,000 times, and a histogram of results was created to obtain statistical behavior (shown in Figure 1). As reported in Table 1, the average MSE of the annual retreat from the randomly generated model was 16.36 m2, as compared to the ArcticBeachv1.0 model value of 125.48 m2 (shown as the red line superimposed on the histogram). The error is much larger for the ArcticBeachv1.0 model than the randomly generated model, while also lying *significantly* outside of the 1st standard deviation of the randomly generated model's "predictions" (predictions in quotations because they are not truly predictions but random numbers). This suggests that the ArcticBeachv1.0 model has

predictive skill, but its predictive skill is *opposite* of the observations (in the direction of larger error or in the opposite direction from mean annual retreat). This is clearly seen by inspection of Figure 4a in the manuscript, where large mismatches in the opposite direction from the mean annual retreat rates are predicted by the ArcticBeachv1.0 model, especially between years 2002 - 2018.

[Figure 1 is given in original CC1].

The test was repeated for the Drew Point location. For the DP location, a random number between a plausible range of retreat was generated for each year using Python's random package. The range in retreat was calculated as the minimum and maximum of the observed retreat data with a 10% envelope (e.g. 5.94 m – 24.83 m). The MSE for the randomly chosen annual retreat was calculated against the observations. This numerical experiment was performed 5,000 times, and a histogram of results was created to obtain statistical behavior (shown in Figure 2). As reported in Table 1, the average MSE of the annual retreat from the randomly generated model was 53.85 m2, as compared to the ArcticBeachv1.0 model value of 61.55 m2 (shown as the red line superimposed on the histogram). In this case, the ArcticBeachv1.0 model performed slightly worse than the randomly generated model (since the MSE for the ArcticBeachv1.0 model was higher than the mean MSE for the randomly generated model). Additionally, the MSE for the ArcticBeachv1.0 model sits within the 1st standard deviation of the MSE for the randomly generated model. This suggests that the ArcticBeachv1.0 model does not predict erosion rates *significantly* different than a randomly generated number. If it did, then the MSE would be well below the 1st standard deviation of the randomly generated model. The performance can also be seen by inspection of Figure 4b, where the ArcticBeachv1.0 model predictions end up under- or over-estimating the retreat at DP, in the OPPOSITE direction just as many times as they estimate the retreat in the CORRECT direction (above or below the mean retreat over the time period).

[Figure 2 is given in original CC1].

Furthermore, the analysis was extended to quantify the error in the cumulative erosion. The cumulative erosion error was calculated as the difference between the sum of the observed annual retreat values and the sum of the modeled annual retreat values. As reported in Table 1, the cumulative retreat error was 48.77 m (reported as "roughly 40 m" in the manuscript text, line 215) for the MK location, and 3.42 m (reported as "within a few meters" in the manuscript text, line 215) for the DP location.
Similarly to the random model numerical experiments presented for the annual retreat predictions, the same procedure is repeated for the cumulative erosion error. For each year, using the same set of random numbers that were generated for annual retreat, the cumulative retreat was calculated by summing the random annual retreat values for each numerical experiment. A histogram was created for each location, shown in Figure 3 (MK) and Figure 4 (DP).
For the MK location, the mean cumulative erosion error for randomly generated model was 39.19 m (see Table 1), as compared to the ArcticBeachv1.0 model value of 48.77 m2 (shown as the red line superimposed on the histogram). In this case, the ArcticBeachv1.0 model performed slightly worse than the randomly generated model (since the cumulative erosion error for the ArcticBeachv1.0 model was higher than the error in the randomly generated model). Moreover, the mean cumulative erosion error for the ArcticBeachv1.0 model sits within the first standard deviation of the cumulative erosion error for the randomly generated model. This suggests that the ArcticBeachv1.0 model does not predict cumulative erosion

*significantly* different than a randomly generated number at the MK location. Interestingly, while the annual retreat predictions were skillful (albeit in the opposite direction), the cumulative retreat might as well have been generated at random.

[Figure 3 is given in original CC1].

At the DP location, the mean cumulative erosion error for randomly generated model was 20.28 m (see Table 1), as compared to the ArcticBeachv1.0 model value of 3.42 m2 (shown as the red line superimposed on the histogram). In this case, the ArcticBeachv1.0 model performed significantly better than the randomly generated model (since the cumulative erosion error for the ArcticBeachv1.0 model was lower than the error in the randomly generated model and it was positioned outside of the 1st standard deviation of the randomly generated model error). This makes sense because the ArcticBeachv1.0 model did a decent job predicting the erosion rate at Drew Point for years in which the erosion was relatively average, but happened to over- or under- estimate the erosion for anomalous years at roughly equal magnitudes, and as a result summing to roughly zero, thus providing little contribution to the cumulative retreat error metric.

[Figure 4 is given in original CC1].

The conclusion from the analysis for predictive skill shows that the ArcticBeachv1.0 model has no predictive skill at the DP location, and has inverse predictive skill at the MK location. Based on the error analysis, I disagree with the authors, as stated in the abstract, that the ArcticBeachv1.0 model provides a promising starting point to project the retreat of Arctic shorelines, or to evaluate historical retreat in places that have had few observations. The results of this analysis at both locations indicate that the model in its current form is under-developed, and cannot be relied upon to provide robust and skillful predictions for coastal retreat rates in the Arctic more than a randomly generated number can (in the case of the DP location) nor can be relied to provide a prediction in the correct trend direction (in the case of the MK location). As disheartening as this error analysis seems, the MK location does show promise because of its ability to capture opposite trends. I suggest to the authors to investigate this behavior more closely, as it probably indicates some physical behavior captured in the model that may be relevant for erosion rates, but in the opposite sense.
Table 1: From what I understand, the two study locations have identical material properties, but they differ in geometry only with cliff height. Is this an adequate demonstration of the model's ability to provide a "physics-based numerical model that can be applied across all partially frozen shorelines"? (Quote from lines 36-37) I was expecting more diversity between demonstration sites.

We thank Jennifer Frederick for her perspective, and realize we needed to add more description in the manuscript that make clear how unambiguously different our validation sites are. We have now added a new subsection called 'Validation Sites' that describes them in more detail. However, we would also like to point out that one should not look at the simplified material properties that represent study sites in a model framework in order to get an idea of what that study site is like in real life. When there is no effort to examine the differences between the real-life sites themselves, one will have a misconception of each site because simplifications must be taken when representing a real-life site in a model. This misconception is exemplified by the approach taken by Jennifer Frederick in her comment above when she states that the 'two study locations have identical material properties, but they differ in geometry only with cliff height' and there is not enough 'diversity between demonstration sites.' We note that it is impossible to capture all of the material diversity at

the spatial and temporal scale of real-life into a model, that our values come from published literature (see references in Table 1) and we would like to highlight that one should not look at model representations of study sites to judge what the real-life site is like.

When we want to examine if physical segments of Arctic coastlines are different, we go directly to the coast itself during fieldwork (as has been done for many years by co-authors of this work, with a decades-long strong German-Russian-Canadian collaboration). We also examine published observational literature (naturally, including those papers the authors have not written themselves, because these sites have been the focus of co-authors work), and also examine historical and satellite data.

In addition, not only are the real-life physical coastline properties just a part of the story of what determines differences at a coast, but what also must be included are the differences in the climate variables at the coast, such as sea ice coverage and wind speed and direction. All of these parts of the story are vital to understand if one would like to compare whether or not two coastlines are similar. Our study sites, for example, Mamontovy Khayata, on Bykovsky Peninsula, Siberia, and Drew Point, Alaska, USA, are starkly different in their main erosional features (e.g. dominance of block erosion at Drew Point, mentioned in the manuscript on lines 32-34 and also in newly added Methods subsection 2.1.3, see response to Reviewer #1) and Mamontovy Khayata (e.g. dominance of thermodenudation, now highlighted in the new Methods subsection 2.1.3 in response to Reviewer #1).

We would also like to point out that our two study sites are located on roughly opposite sides of the Arctic Ocean, they have different open water season lengths (Figures 6-8), wind speed and direction (Figures 6-7) and bathymetry. These differences are all taken into account in our coupled storm surge arctic erosion model, and consequently in our coupled erosion model. This is mentioned in Section 2, lines 58-67.

**References:**

Are, F. E.(1988). Thermal abrasion of sea coasts, *Polar Geography and Geology*, 12, 1–86, from: Termoabraziya morskikh beregov, Nauka, Moscow, 1980, 158 pp. doi:10.1080/10889378809377343

Casas-Prat, M., & Wang, X. L. (2020). Projections of extreme ocean waves in the Arctic and potential implications for coastal inundation and erosion. *Journal of Geophysical Research: Oceans*, *125*(8), e2019JC015745.

Günther, F., Overduin, P. P., Sandakov, A. V., Grosse, G., & Grigoriev, M. N. (2013). Short- and long-term thermo-erosion of ice-rich permafrost coasts in the Laptev Sea region. *Biogeosciences*, *10*(6), 4297-4318.

Kobayashi, N., Vidrine, J. C., Nairn, R. B., & Soloman, S. M. (1999). Erosion of frozen cliffs due to storm surge on Beaufort Sea Coast. *Journal of coastal research*, 332-344.

Jones, B. M., Hinkel, K. M., Arp, C. D., & Eisner, W. R. (2008). Modern erosion rates and loss of coastal features and sites, Beaufort Sea coastline, Alaska. *Arctic*, 361-372.

Jones, B. M., Arp, C. D., Jorgenson, M. T., Hinkel, K. M., Schmutz, J. A., & Flint, P. L. (2009). Increase in the rate and uniformity of coastline erosion in Arctic Alaska. Geophysical Research Letters, 36(3).

Jones, B. M., Farquharson, L. M., Baughman, C. A., Buzard, R. M., Arp, C. D., Grosse, G., ... & Romanovsky, V. E. (2018). A decade of remotely sensed observations highlight complex processes linked to coastal permafrost bluff erosion in the Arctic. *Environmental Research Letters*, *13*(11), 115001.

Lantuit, H., & Pollard, W. H. (2008). Fifty years of coastal erosion and retrogressive thaw slump activity on Herschel Island, southern Beaufort Sea, Yukon Territory, Canada. *Geomorphology*, *95*(1-2), 84-102.

Lantuit, H., Atkinson, D., Paul Overduin, P., Grigoriev, M., Rachold, V., Grosse, G., & Hubberten, H. W. (2011). Coastal erosion dynamics on the permafrost-dominated Bykovsky Peninsula, north Siberia, 1951–2006. *Polar Research*, *30*(1), 7341.

Lantuit, H., Overduin, P. P., Couture, N., Wetterich, S., Aré, F., Atkinson, D., ... & Vasiliev, A. (2012). The Arctic coastal dynamics database: a new classification scheme and statistics on Arctic permafrost coastlines. *Estuaries and Coasts*, *35*(2), 383-400.

---

## Author Comment (AC2)

**Author Response to Reviewer #1.**

**The comments in by Reviewer #1 are in font color black. The authors' responses are in green. The changes to the revised manuscript are in blue.**

*Geoscientific Model Development* gmd-2021-28
**1** of **1**

**Summary**
Thank you for the opportunity to conduct a review of this manuscript. Here, the authors couple an existing storm surge and erosion model to estimate annual rates of coastal erosion for two study areas in the Arctic (Drew Point, AK and Manmontovy Khayata, Siberia). The authors conclude that they can predict multi-annual cumulative erosion on the same order of magnitude of what has been observed at these sites and that their methodology is an important first-step toward an approach for estimating erosion for pan-Arctic scales.

**Recommendation**
I commend the authors on their writing styles, as evidenced by the small number typographical errors throughout the manuscript. However, this work hosts a multitude of technical issues, most notably the study's methodology and conclusion based therefrom. My feeling is that it does not warrant publication. For this reason, I have limited my review to two major comments, as opposed to more detailed in-line comments.

We thank the reviewer for their perspective, and provide detailed explanations below addressing the statements by the reviewer.

**Major Comments**
The authors highlight that "the most important root causes of Arctic shoreline change can only be gained through careful evaluation of the physical processes involved" and yet make no such effort for their own study. For example, one of the two sites where the authors apply their model is Drew Point, AK. Here, it is well known that permafrost blocks bound by ice wedges topple onto the beach due to an undercutting process that is facilitated by storm surge (i.e., "thermo-abrasion"). This reality is in stark contrast with the incremental style of bluff retreat associated with the model of coastal erosion employed by the authors (Figure 1). I understand that the authors ultimately wish to exercise their modeling framework elsewhere, but what is the scientific value of applying such a model to a place like Drew Point? My feeling is that Drew Point is not an appropriate location to apply or test the erosion model the authors use in this study.

We appreciate this comment by the reviewer and acknowledge that the purpose of choosing Drew Point as one of the validations sites for our study was not described clearly enough in the manuscript. The reason we have chosen Drew Point as a validation site for our model is exactly *because of* the special block erosion processes there, and the fact that ArcticBeach v1.0 purposefully does not include a specific representation of block erosion or any other complex three-dimensional process. Instead, our model aims to approximate coastal erosion as dimensional diffusive processes including a tunable bulk (offset) parameter accounting for unknown or unrepresented processes. To further elucidate the reasoning behind this decision, we provide the following main points:

- Block erosion is not the typical erosive process considering the arctic shoreline as a whole (Lantuit et. al. 2011, also stated in line 34 of the manuscript). Our approach is

to include those physical processes that are most important in driving coastline retreat that can be applied across the whole arctic coastline, and not just one short segment like Drew Point, Alaska.

- We do not aim, at this point, to add physical processes that are only specific to certain stretches of coastline. We find this contrary to our goal of providing an order-of-magnitude estimate of erosion rates for coastlines that erode with different dominating processes. Through this approach, we take a first step for a physical parameterization that is well-suited to be incorporated into a larger earth system or coupled model, without having to resolve differences in specialized processes for certain coastline segments. This approach is justified at lines 30-38 in the manuscript.

- So, in order to validate if our model would be useful for simulating erosion on a coastline with a special process such as block erosion, when our model does not include block erosion itself (indicated in Section 2, line 66 of the manuscript), we carefully selected Drew Point, where block erosion is a special, dominant feature.

.

- While we did address coastline-specific processes not included in the model in Section 4.2.1, lines 370-379, and addressed the reason we chose to leave some (such as notch erosion) out (Section 1, lines 32-34 and lines 36-37) and instead indirectly calculated (see response to Reviewer #2), we can see that it should have been more clearly explained in the manuscript, and have added what has been changed in the revised manuscript below in italics (now a new Section in the revised manuscript, Section 2.1.3, 'Validation Sites').

- We also understand that in the field of arctic erosion, many scientists have a research focus on the extensively-documented and special-case processes occurring at the rapidly eroding Drew Point (several of these models are referenced in the manuscript in Section 1, lines 27-30), and therefore might be concerned with our conscious choice to not include block erosion in our model. Therefore, we have added detailed statements in a new subsection 2.1.3 more clearly outlining the reasons we chose Drew Point as a validation site.

- We would also like to point out that while our model does not include block erosion, it does provide acceptable estimates of cumulative retreat at Drew Point, within 3.3 m from 2007-2016 (Figure 4). At the same time, the same model can also realistically approximate the cumulative retreat at a site far across the Arctic Ocean, characterized by very different coastal conditions and erosional processes and where block erosion is not a main mechanism (but instead primarily thermodenudation) is what controls coastline retreat (Mamontovy Khayata, Siberia, Figure 4).

- The new subsection (in the tracked changes version of the manuscript: Section 2.1.3, lines 145-162):

*"The validation sites for ArcticBeach v1.0 are Mamontovy Khayata (MK), Bykovsky Peninsula, Siberia and Drew Point (DP), Alaska, USA (Figure 3). These sites were chosen because they: 1) involve specialized processes that are, at this time, purposely excluded in ArcticBeach v1.0, and 2) are coastline segments that are very different from each other. We chose not to include the specialized processes of either DP and MK in our simple model because our goal is to establish one general numerical model that represents a first step at simulating diverse types of Arctic coastline, efficient enough to be incorporated into a greater*

*earth system model. So, to establish this initial model v1.0, we chose these specialized places of MK and DP in order to test whether or not our model could simulate observed retreat, while, at the same time, not including all of the associated special site-specific processes.*

*The differences between the validation sites are highlighted by two main aspects. Firstly, the validation sites differ from each other in terms of their primary erosional processes. At MK, the primary mechanism for erosion is sub-aerial erosion, thermodenudation, and thaw slumping (Overduin et al., 2016; Günther et al., 2015). Coastline retreat at DP, on the other hand, is strongly driven by block erosion (Jones et al., 2018; Ravens et al., 2012). The block erosion occurring at DP is a specialized process that only occurs on very short stretches of Arctic coastline compared to the Arctic coastline as a whole (Lantuit et al., 2012). A second reason these validation sites are so different is that they are physically located far away from each other, such that the environmental forcing (sea ice cover, winds, sea surface temperature) are pointedly different. This allows for the model framework of ArcticBeach v1.0 to be tested because it does incorporate all of these forcing variables (which are also readily available from CMIP model output (Meehl et al., 2000) and reanalysis datasets). In this case, these variables were taken from reanalysis data mentioned in Section 2.3."*

We have also added new statements to Section 4.2.1, Lines 432-437 of the tracked changes version of the manuscript, (please see our response to Reviewer 2):

*"Further, our goal is not to explicitly represent some site-specific processes such as notch erosion, but rather indirectly calculate the effects of seawater on retreat by using Equation 1. This approach leaves the opportunity to utilize ArcticBeach v1.0 on a range of coastlines that have different erosional processes which do not include notch erosion as a primary mechanism for retreat (see Section 2.1.3). Notch erosion is thus indirectly calculated in Equation 1 with the terms $d_c$ (water depth at the cliff toe, which must be positive for the erosion module to be activated, see also Figure 1) and $l_c$ which refers to the length of cliff exposed to the seawater."*

My biggest concern regarding the validity of this study is the lack of an error analysis of the model outputs (i.e., annual rates of erosion). The model predictions are higher and lower than the observations and in a somewhat chaotic fashion (Figure 4a-b). In many cases, the model predictions are several factors (approaching an order of magnitude) off. Given that the calibration of the model includes an input of historical retreat rates, is this level of error acceptable? What explains the seemingly non-systematic trends in model error?

The reviewer raises the question of what causes the error and 'seemingly non-systematic' trends in modelled retreat rates compared to the observed retreat rates (Figure 4a-b). We felt that addressing this behavior was so important that it warranted its own subsection in the manuscript (Discussion section 4.1), but since the reviewer has still raised this question, we see that this section must be addressed more clearly. Essentially, what we had tried to explain in Section 4.1 is that due to the fact that the tuning parameter used in the model is the mean of a timeseries of annually-calibrated values, and the difference between this mean value and a given annual value will directly determine if the model will over- or under-estimate erosion for that year. For example, years when the annual tuning parameter values are above the timeseries median (in other words, the years where the red stars in Fig. 8 are above the red dashed line) are the same years when the model underestimates annual retreat rates (the years where the blue bar is lower than the orange bar in Figures 4a and b).

For someone just quickly skimming through the figures, we have also added statements to the captions of both Figures 5 (previously Figure 4) and Figure 9 (previously Figure 8), so that this *purely systematic* variability is explained *directly in the figures*, instead of the reader having to dive in to the relevant Discussion Subsections 4.1 and 4.1.1.  This way, we hope we have now made it clearer that the 'seemingly non-systematic trends in model error' are, in fact, very systematic, and can be directly explained by the way we have performed the model calibration.

*Please see also our very relevant changes to the manuscript in response to Reviewer 2 comments Section [c] and Reviewer 2 Section: 'Results'.*

New caption to Figure 5 in the revised manuscript (previously Figure 4) that refers to Figure 9 (previously Figure 8):

"*The years when the observed retreat rates are under(over)-estimated are the same years when the annual values of the so-called 'water level offset', a proxy for the physical processes at this point unresolved by the model, are above(below) the median values.  These years are indicated where the red star is above(below) the red dashed line in Figure 9.*"

New caption to Figure 9 in the revised manuscript (previously Figure 8) that refers to Figure 5 (previously Figure 4):

*When the annual water level offsets (red stars) exceed the median water level offset (red dashed line), the model predictably underestimates observed retreat rates (see corresponding years in Figure 5) and vice versa.*

We have also added these statements to Section 2, lines 70-74 of the tracked changes version of the manuscript:

"*Small scale processes such as niche formation are accounted for in a bulk tuning parameter (Section 2.5) in this coarse-scale approach. We would like to point out that the model is not aiming for reproducing individual years and erosional events at a specific point, but to deliver large spatial scale and long term (decadal) approximations of coastal erosion related to the physical environmental conditions. This is also the reason why we restricted model tuning to only a single offset parameter.*"

**I calculated a negative value for the Nash-Sutcliffe Model Efficiency (EF) statistic using the measured vs. modeled erosion rates reported for Drew Point in this study, which indicates that the mean of the Drew Point observations is a better predictor of annual erosion than the author's model. This back of the envelope calculation with a widely used error analysis metric underscores a potentially major issue regarding the predictive power of the author's model.**

The EF is given by:

$$EF = \left[ \sum_{i=1}^{n} \left( O_i - \overline{O} \right)^2 - \sum_{i=1}^{n} \left( P_i - O_i \right)^2 \right] \Big/ \sum_{i=1}^{n} \left( O_i - \overline{O} \right)^2 ,$$

where $P_i$ are the predicted values, $O_i$ are the observed values, n is the number of samples, and is the mean of the observed data. The EF statistic ranges from 1.0 to $-\infty$, with 1.0 indicating a perfect match between $P_i$ and $O_i$ and EF less than zero indicating that is a better model than $P_i$ for simulating $O_i$.

Without a formal error analysis or comparison to another erosion model, it difficult to argue that this study has advanced our understanding of Arctic coastal erosion processes or produced meaningful insights for the communities that are vulnerable to this environmental problem.

We thank the reviewer for conducting this analysis. We understand that while the mean of the observed erosion rates may be a good predictor of past erosion rates, we would like to highlight that the Arctic (thus the controlling environmental variables such as open water duration and temperature) is changing in a non-linear fashion, thus the need for physics-based numerical models becomes urgent to understand what sort of changes we will expect to see in the future. It is not possible to use the EF analysis above on future erosion rates because we do not have observations of the future. We argue that while ArcticBeach v1.0 indeed does not perfectly reproduce past erosion rates at specific points and individual years, it does simulate realistic orders of magnitude (Figure 4), and, just as importantly, provides a framework for projecting retreat rates accounting for transient environmental conditions. Projected wind forcing is available, for example, from CMIP models, which have been built from well-known geophysical principles, and from wind speed and direction, the coupled storm surge model in ArcticBeach v1.0 can calculate what sort of relative water levels we will have in the future. Coastal retreat is only one part of ArcticBeach v1.0, and the other part is calculating relative water levels at the coastline where bathymetry is only roughly known (Section 2.2). Water levels at the coastline are an essential driving process of coastline erosion, and it is therefore useful to have such a water level model incorporated into the erosion model. The water levels calculated even take into account varying periods of sea ice cover, as described by reanalysis data. Relative water levels were able to be reproduced well (RMSE values given in the newly added Table 2), as described in Section 3.2 and shown in Figure 5 (now Figure 6 of the revised manuscript). We have also provided RMSE values in a newly added Table 2 (see response to Reviewer 2) for retreat rates.

In addition, processes impacted by projected increases in temperature, such as accelerated erosion due to thermodenudation and sea level rise, can also be factored into ArcticBeach in future work. This is discussed in Section 4.2.1, lines 371-379, and Section 4.3 (Please see the additions to these subsections in the revised manuscript as mentioned in response to Reviewer 2).

In summary, we should expect these coupled and non-linear processes will cause future erosion rates to deviate from the mean of past observations also in a nonlinear fashion. Therefore, using the mean erosion rates today will likely not be the best way to predict erosion rates into the future, and forcing variables such as water level must be taken into account in addition to other physical processes also present in our model (Sections 2.1 through 2.1.2, and newly added Figure 1 (please see response to Reviewer 2). While we have stated in the Introduction that 'statistical methods might therefore show promising results to simulate erosion rates' at line 25-26, but in response to the reviewer's comment, we have also added statements in the Introduction section to highlight and further explain the need for such

a numerical model and why we cannot rely solely on the mean of past retreat rates to predict future rates.

Added to Section 1, Lines 26-30 of the tracked-changes revised manuscript:

*"Further, current statistical relationships of coastal erosion to other climate variables will change in the future because changes in the Arctic are happening in a non-linear fashion. In addition, how tightly certain environmental processes are coupled to erosion is also changing. For example, wave action in the Arctic is increasing nonlinearly, leading to variability of how vulnerable Arctic coastlines are to erosion in the future (Casas-Prat and Wang, 2020)."*

---

## Author Comment (AC3)

**Author Response to Reviewer #2.**

**The comments in by Reviewer #2 are in font color black. The authors' responses are in green. The changes to the revised manuscript are in blue. The blue line numbers refer to the section and line numbers in the tracked changes version of the revised manuscript.**

**A Novel Approach**

Anonymous Referee #2

Referee comment on "ArcticBeach v1.0: A physics-based parameterization of pan-Arctic coastline erosion" by Rebecca Rolph et al., Geosci. Model Dev. Discuss., https://doi.org/10.5194/gmd-2021-28-RC2, 2021

**General description of the paper**
First of all, thank you for allowing me to review the paper. The paper was well written, the problem statement and the solutions are explained in detail. The writers developed a simplified model for large scale modelling despite limited available measurements of the parameters. The authors coupled basically three major numerical modules with different physical processes like cliff and beach erosion with storm surge interactively. The models, albeit simplified, are based on real-world physics. The authors used mainly water level to calibrate the model. The other inputs of the forcing parameters like wind speed, wind temperature and water temperatures were taken from global models.

We sincerely thank the reviewer for his/her feedback on our manuscript, and appreciate the concise description above which we consider to be a very accurate summary of our approach and manuscript.

**Major comments**

**Technical issues**

[a] Uniform statistical distribution is used for sensitivity analysis. In Table: 1, a range of the most influential parameters are provided. The range for each environmental parameter is quite broad. Justification to apply uniform distribution is under question. Did the authors try any other distributions with central tendency?

We agree that the applied uniform distribution appears to be an arbitrary selection in the previous version of the manuscript. However, we argue that there is very limited range of observations available for the parameters examined (ground ice, sediment size, cliff height, and the other parameters listed in Table 1). Given that the prior probability distributions of these parameters are unknown, any other distribution would be arbitrary as well. A uniform distribution, however, avoids underestimation of parameter values close the assumed parameter limits. A statement justifying the use of a uniform distribution has been added:

Section 2.6, Lines 241-242: *"We chose a uniform rather than a central distribution because it provides a more comprehensive assessment of error, given that observations are relatively few and so we cannot confidently assess prior probability distributions."*

[b] The authors explained the effect and importance of the 'offset water level' as a proxy for some excluded physical process. Section 4.2.1 might be the place where it may be explained how water level offset indirectly compensates or estimates the notch erosion mechanism [authors did mention that the process is excluded in line 66 and also in Section#1 citing the notch erosion mechanism is not so common] Was equation#1 used to indirectly calculate notch erosion since the equation covers the portion of the cliff that is in contact with warmer seawater? This can be one explanation of why the model works despite excluding the block failure by the wave-created-notch mechanism.

Yes, we appreciate this comment and have now added the following statements to the manuscript that further elucidate and highlight our approach, which the reviewer has correctly explained above:

Section 4.2.1, Lines 432-437: *"Further, our goal is not to explicitly represent some site-specific processes such as notch erosion, but rather indirectly calculate the effects of seawater on retreat by using Equation 1. This approach leaves the opportunity to utilize ArcticBeach v1.0 on a range of coastlines that have different erosional processes which do not include notch erosion as a primary mechanism for retreat (see Section 2.1.3). Notch erosion is thus indirectly calculated in Equation 1 with the terms $d_c$ (water depth at the cliff toe, which must be positive for the erosion module to be activated, see also Figure 1) and $l_c$ which refers to the length of cliff exposed to the seawater."*

[c] Assessment of how the model is performing should be determined. As a proof of concept, the model makes a strong argument. However, the accuracy of the validation is still warranted.

We agree and we have now calculated the root mean square error of both the erosion module and the storm surge module. These have been added in a new table in the text and also in their respective places in the text:

Section 3.1, Lines 271:-272 *The root mean square error (RMSE) of simulated coastline retreat for MK is 7.84 m and 7.23 m for DP (Table 2).*

| Coastline retreat [m] | Water level [m] |
|---|---|
| 7.84 (MK) | 0.35 (MK) |
| 7.23 (DP) | 0.16 (Prudhoe Bay) |

*Table 2. The root mean square error (RMSE) of simulated coastline retreat and water levels for the study sites. At DP, no observed water levels are available, so the water levels from the nearby tide gauge at Prudhoe Bay were used, as described in Section 2.4. Prior to calculating the RMSE of modelled water levels at Prudhoe Bay, the mean offset between the modelled and observed water level was first removed because the water level observations and water level model correspond different baselines (see Section 2.5).*

Section 3.2, Lines 292-293: *"The RMSE for the storm surge model at the MK is 0.35 m. For Prudhoe Bay, the RMSE was calculated after removing the mean offset caused by a different relative baselines described above and was found to be 0.16 m (Table 2)."*

Added to caption of Figure 6: *"… relative to a theoretical still water depth…"*

We have also added the following statements that clarify when an over- or underestimation of modelled retreat occurs, due to our calibration setup:

Section 2.5, Lines 234-237: *"When the annual water level offset exceeds the median of the entire water level offset timeseries, it follows that the modelled retreat will be underestimated for that year, and vice versa. This is due to the calibrated summed water level that is applied to simulate erosion being lower than the annual water level necessary to reproduce the exact erosion rate for the given year."*

Section 3.1, Lines 259-260: *"This over- and underestimation is expected when we examine the annual water level offset values in comparison with the median water level offset value that was used in model calibration (Section 2.5)."*

Added to the caption of Figure 5 (previously Figure 4): *"The years when the observed retreat rates are under(over)-estimated are the same years when the annual values of the so-called 'water level offset', a proxy for the physical processes at this point unresolved by the model, are above(below) the median values. These years are indicated where the red star is above(below) the red dashed line in Figure 9 (previously Figure 8)."*

Added to the caption of Figure 9 (previously Figure 8): *"When the annual water level offsets (red stars) exceed the median water level offset (red dashed line), the model predictably underestimates observed retreat rates (see corresponding years in Figure 5 (previously Figure 4) and vice versa)."*

[d] A flow chart may be included in 'Chapter#2: Methods' to describe the methodology concisely. For example, it is not clear from the descriptions when and where the erosion process was 'not simulated' in the model. As understood, two binary switches (on/off) exist in the model: (1) the open water season in the time domain and (2) collapsed but not-yet-eroded sediments on the beach in the space domain.

We thank the reviewer for this suggestion. A flow chart has been added, and we feel that this comment has greatly improved the methodology section of the paper. The reviewer also had those two statements above correct, but to prevent any possible doubt by the reader, a flow chart becomes necessary. This figure has been added to Section 2: Methods of the manuscript:

[Figure]

*Figure 1. A conceptual flow chart summarizing the main inputs (purple) and processes (grey) of ArcticBeach v1.0. Climate forcing and rough bathymetry are used to drive a storm surge module (Freeman et al., 1957). The resulting water levels are then used to drive the erosion module (Kobayashi et al., 1999). A schematic of the erosion module is given in Figure 2. Under times of sea ice cover at the coast (assumed when sea ice concentration exceeds 15%), erosion is assumed to be negligible and neither module is activated.*

**Comments in general**

Introduction

The introduction is well written. The requirement to establish a pan-Arctic level model is explained. The authors explained sufficiently the requirement of a simple physics-based model and the benefits of such a computationally inexpensive model.

We are glad that the reviewer thinks our introduction is well written and are also delighted to see that he/she sees the need for a computationally-inexpensive numerical model to simulate Arctic coastal erosion.

Methods

The conceptual models are explained in this section. The major numerical modules are erosion module comprising cliff and beach erosion based on thermal energy transfer from water to the cliff via convection and a quasi-steady storm surge model based on wind speed. The conductive heat transfer and solar radiation are not included in the model. The authors did not provide the explanation of excluding the other two heat transfer mechanism but it is reasonable to assume, the *solar radiation* is indirectly included in the seawater temperature inputs, whereas the effect of the *conduction* is 'felt' as time-lag which can be ignored when modelled for a long duration.

We appreciate that the reviewer has pointed out the two other heat transfer mechanisms of solar radiation and conduction. We also confirm that, indeed, we chose not to add to these in a new form in the Kobayashi et al. (1999) model (the erosion module of ArcticBeach v1.0). We have now added the following explanation to the manuscript:

Section 2.1.2, Lines 139-143: *"Consistent with the chosen erosion module in ArcticBeach v1.0, Kobayashi et al. (1999), conductive heat transfer and solar radiation are not directly included. Solar radiation can be partially accounted for in the sea surface temperature input and sea ice cover (see Section 2.3). Conduction effects are much smaller than effects of solar radiation over long time periods and are neglected. However, the opportunity to include effects of solar radiation can be implemented in later versions of the model, to include processes such as thaw slumping and 1-D heat-transfer permafrost models as described in Section 4.2.1."*

The authors correctly identified the problem of determining absolute water level at the toe of the cliffs and provided the detailed methodology of circumventing the issue and reaching a reasonable solution. A small description of the statistical method of Monte Carlo is also provided which might be elongated.

Yes, we have elongated the Monte Carlo section by now providing an example which we hope further clarifies our approach:

Section 2.6, Lines 246-250: *"To further illustrate our Monte Carlo method, we will use the example of how changes within a uniform distribution of observed ice content can be expected to change the modelled retreat rates. We ran ArcticBeach v1.0 a total of 500 times for each site, and for each model run, a certain percentage of cliff ice was assigned to a different value each time but within the observed range of 60-90% (given in Table 1). In this example, since all other parameters remained unchanged except ice content, this resulted in a distribution of retreat rates caused by changes in cliff ice content."*

Results

Results are discussed by comparing the outputs of the model with the observations. However, the estimation of the accuracy is not determined. One of the model outcome anomalies is the underestimate of the erosion from 2002 to 2009 is identified, but authors need to provide a strong explanation of the deviation.

We greatly appreciate this comment that the accuracy estimation was not clear enough in the manuscript. We would also like to refer our answer to this reviewer's related comment in letter [c] above, which explains how we addressed the validation of the model in the Results section.

Besides the other changes responding to letter [c] above, we made sure to explicitly mention the underestimation of erosion from 2002-2008:

Section 3.1, Lines 264-272: *"To further illustrate how we can expect when the model will over or underestimate observed retreat, we will take the example of the underestimation of coastline retreat at MK during the period of 2002-2008 (Figure 5a). This underestimation of retreat is caused by the annual water level offsets calculated for 2002-2008 being above the median water level offset used in the model forcing (see red stars above the red dashed line for 2002-2008 in Figure 9a* [previously Figure 8a]*). This means that the calibrated water*

*level required to reproduce the observed retreat for 2002-2008 is higher than the median of the calibrated water level to reproduce the observed retreat across the entire timeseries. While bulk calibration inevitably leads to errors for individual years, we find this approach is still able to capture cumulative retreat over a long timeseries well (Figure 5c,d)."*

**Grammar and Comprehension**

The script is admirably laid out. It is recommended to re-write very few sentences ( marked in the attached pdf)

We are glad that the reviewer found the manuscript easy to navigate, but also appreciate the grammatical corrections provided in the supplement. They have all now been implemented.

Abstract, Line 2: 'This change in climate…' to *'Climate change...'*

Line 43: 'than' to *'from'*

Line 191: 'come' to '*comes'*

Line 254: 'given to '*shown'*

Line 419-420: *'…on erosion rates than at MK because …'* instead of '…of erosion rates than at MK.  This is due to …'
* * *
**Recommendation**
The journal paper is recommended to publish with minor modifications. The work provides a novel approach to simulate coastal erosion. This is one of the early efforts to understand Arctic coastal erosion on a global level. The authors chose to use simplified models in favour of lower computation expenses and it is reasonable to exclude some physical processes. The novelty of the work is the coupling of the modules, calibration of the coupled model with water level and application of the model in two different sites. Please also note the supplement to this comment:
https://gmd.copernicus.org/preprints/gmd-2021-28/gmd-2021-28-RC2-supplement.pdf

---

## Author Comment (AC4)

**The comments by Reviewer #3, Anatoly Sinitsyn, are in font color black.** **The authors' responses are in green. **The changes to the revised manuscript are in blue. The line numbers refer to the section and line numbers in the tracked changes version of the revised manuscript after addressing the comments from Reviewer 3.**

Geosci. Model Dev. Discuss., referee comment RC3
https://doi.org/10.5194/gmd-2021-28-RC3, 2021
Comment on gmd-2021-28
Anatoly Sinitsyn (Referee)

**Referee comment on "ArcticBeach v1.0: A physics-based parameterization of pan-Arctic coastline erosion" by Rebecca Rolph et al., Geosci. Model Dev. Discuss., https://doi.org/10.5194/gmd-2021-28-RC3, 2021**

*Anatoly Sinitsyn,*
*June 27 2021, Trondheim*

Review of the article "ArcticBeach v1.0: A physics-based parametrization of pan-Arctic coastal erosion" by Rolph et al. (2021)

The article presents a model for estimation of coastal dynamics at permafrost, ice-rich coastlines. More specifically, erosion rates at coastal bluff and beach are handled by the model. The model utilizes 1-D coastline erosion model of Kobayashi et al. (1999), bathystrophic storm surge model of Freeman et al. (1957), and empirical equations of Kriebel and Dean (1985) for estimating cross-shore sediment transport. The model is forced by historic hydrometeorological data (wind speed and sea ice concentration), and initialized by existing bathymetry of the case study locations. The model is validated by observed water level data. Sensitivity of the modelled retreat rates is accessed with the Monte Carlo approach. Modelled retreat rates are compared with observed rates for evaluation of the model performance. It was found that the water level plays critical role in defining retreat rates. The results demonstrate that the model is capable to reproduce retreat rates withing the same order of magnitude as the observed retreat rates. This is promising result justifying the model performance, and possibilities of application for crude assessments of coastal dynamics in relevant coastal settings.

The model developed by the authors looks definitely useful for the field of arctic coastal dynamics, and shall be considered as a very good step forward.

I have several, largely suggestive comments, which are presented in the attached file. Intension of these comments is to clarify some points in the text of the article and make it more suitable for engineering community, who is not necessary dealing with permafrost coastlines on a daily basis.

We sincerely thank Anatoly Sinitsyn for taking the time to review our manuscript and greatly appreciate his useful comments. Our responses are below.

Main point of my comments are the following:

- Despite the title, the model is aiming to handle some, but not all, of the morphologies comprising pan-Arctic coastlines, i.e. ice-rich coastal bluffs/coasts. This limitation could be mentioned in the text otherwise the article might provide to a reader a hope on a generic model applicable to all Arctic coastlines, or a vision that the Arctic coasts are all ice-rich.

We understand that the text with its respective title could leave the reader with the impression that all Arctic coasts are ice-rich, and we thank the reviewer for pointing this out. However, we do argue that our model could be applied on ice-poor lithified coastlines with very little modification (e.g. setting ice content to 0% and assigning the appropriate water level calibration). Despite this, we did choose to focus on rapidly retreating coastlines, and since these tend to be ice-rich, unlithified coastlines, we have not yet tested our model on ice-poor or lithified coastal segments. In the revised manuscript, we make it clearer to the reader that this model has not been tested on the other types of coastlines by adding the following statement (and other references to sandy coastlines, please see our responses below).

Abstract: Lines 17-19: "This proof-of-concept model is tested on ice-rich, unlithified coastlines. Through flexibility of input parameter choice (e.g. ice content, cliff height), the framework permits application to ice-poor, lithified or sandy coasts.'

- As continuation of the previous comment, it looks natural, if such modelling attempt would aim to model or refer to a well-described coastal process such as thermal abrasion or thermal denudation, and to model a core component of such processes. If fact model do model components of such processes. This would help to compare the model results with direct field observations. sOne may object that it is just a sense of usage of a certain terminology, as the article is efficiently deals with the processes called thermal abrasion and thermal denudation. Still, due to the aforementioned points, the article looks somewhat detached from the body of literature describing the processes on the Artic coastlines.

Thermal abrasion is directly taken into account when calculating the convective heat transfer coefficient between the wave action of the relatively warmer seawater and the coastline, and it includes wave height, period, and depth (Equations 10-11 in Kobayashi et al. 1999). The convective heat transfer coefficient equation has been added to the revised manuscript in Section 2.1.1, Equation 2. Thermal denudation, on the other hand, is not explicitly taken into account but we have now added an appropriate reference to it in Section 4.2.1, Line 453 when discussing an outlook for the model and coupling of a 1-D surface heat flux model.

Section 2.1.1, Lines 105-118: 'The parameter $h$ is a convective heat transfer coefficient [$\frac{J}{sm^2 °C}$] between the thawing cliff ($h_c$) or beach ($h_b$, Section 2.1.2) surface and warmer seawater. It estimates transfer of heat for a turbulent boundary layer in a unidirectional flow above a flat plate (Schlichting (1968), Kobayashi and Aktan (1986)) and is given by

$$h_{c,b} = \frac{\alpha f_w C_w U_b}{1 + F\sqrt{0.5 f_w}}$$

where α is an empirical parameter included for wave-induced thawing with α = 0.5 for unidirectional flow, $f_w$ is a wave friction factor at the thawing surface that is dependent on equivalent sand roughness of either the cliff or beach, $C_w$ is the volumetric heat capacity of seawater, and $U_b$ is the representative fluid velocity just outside of the boundary layer and takes into account wave height, wave period, and wave depth. $F$ is a parameter that changes according to thresholds imposed on the Reynolds number, which is directly proportional to the shear velocity accompanying the shear stress on the thawing surface, and changes depending on whether there are hydraulically smooth or fully rough conditions. More detailed information on the convective heat transfer coefficient and relevant parameters including $U_b$ and F are provided by Equations 10 and 11 in Kobayashi et al. (1999).

Reference to above equation has been added in Section 2.1.2 Lines 143-144 : '… where $h_b$ is the convective heat transfer coefficient on the exposed frozen beach sediment [J/(s m$^2$ °C)] [and is given by Equation 2].'

Section 4.2.1, Lines, 453 : '…to give a more complete overview of thermal denudation erosional processes at play at permafrost coasts.'

- In motivation for the article, the authors refer to the challenges ice-rich coastlines cause to the infrastructure. It is known from the practice, that it is normal to avoid ice-rich coasts when designing new infrastructural projects. Yet, sometimes handling such coastal type cannot be avoided. Hence, in general terms, relevance of models handling ice-rich sediments for the infrastructure developments might be somewhat limited. Yet, applicability of such models can take place in certain cases with relevant coastal conditions.

Yes, we agree with this comment and thank Anatoly Sinitsyn for bringing up this point. We have made the following changes to the revised manuscript:

Added the word 'existing' to the second sentence in the Abstract: '… causing problems for [existing] industrial, …'

Added statement to the Conclusion section, Lines 515-517. 'Such projected retreat rates from ArcticBeach v1.0 should not be used for infrastructure planning. The model is only capable to deliver first order approximations on how far the coastline will retreat, providing a basis for which associated impacts on already existing infrastructure and nearshore biogeochemistry might be better constrained.'

- As continuation of the previous comment, in my opinion, such model and its further development may consider the needs biogeochemistry on equal footing as the needs of infrastructure.

We also agree with this comment, and would like to direct to our answer in the related comment directly preceding this one. In addition, we have also added to the Discussion:

Section 4.3, Lines 475-480: 'Further development of ArcticBeach v1.0 should consider such biogeochemical applications on an equal or rather higher priority than applications concerning threats to existing infrastructure due to the nature of these two very different applications. Assessing threats to either existing or planned infrastructure generally requires a site-specific model and approach, with very detailed site-specific information and processes. We would like to make it clear that the design of ArcticBeach v1.0 lends itself to more pan-Arctic use

for regional and first-order estimates of retreat rates and associated volume transport of
nutrient-rich sediments into the nearshore zone.'

Sincerely yours,
Anatoly Sinitsyn
Please also note the supplement to this comment:
https://gmd.copernicus.org/preprints/gmd-2021-28/gmd-2021-28-RC3-supplement.pdf
Powered by TCPDF (www.tcpdf.org)

Page numbers refer to the .pdf supplement containing the comments by Anatoly Sinitsyn
(Reviewer 3).

Page 1:

**Commented [AS1]: Comm4**
Yes, coastal erosion is one of the main natural hazards when it comes to infrastructure in the
Arctic. Yet, it is rear when we place the infrastructure at ice-rich coasts, as they are known to
have highest erosion rates. We normally try to place infrastructure at more stable
morphological types, such as barrier islands, river deltas, etc.  If so, then motivation for the
article could be somewhat shifted towards the biogeochemistry.

Yes, we agree and would like to refer to our responses to the two comments directly before
this one.

**Commented [AS2]: Comm1**
Yes, frozen cliff and beach a partially frozen during summer and may be fully frozen during
winter. These are normal variations in the state of a permafrost coast. The model deals with
particular conditions of a permafrost coast. Hence, I would suggest first to outline that the
model is handling a dynamics of a permafrost coast, and then, if the authors think that that is
necessary, to point out that the model focused on partially frozen cliff and beach.

Yes, this is a good point—we have thus changed the following phrases: 'partially frozen cliff
and beach' to 'permafrost coast' in Line 7 of the abstract and also 'partially-frozen' to
'permafrost' in Section 4.2.1, Line 453.

**Commented [AS3]: Comm2**
Arctic coastlines include different morphologies (+ sandy beaches, rocky coasts, river deltas),
not only ice-rich coasts. Is the ice/rich coasts the dominant morphology in the Arctic? Perhaps
such coasts constitute some 40-50 %, then one could say that the model will support such
large-scale models when it comes to unlithified coasts.

We agree this is an important point and we have added a statement to the abstract—please see
our response to the first bullet of Anatoly Sinitsyn's review above.

**Commented [AS4]: Comm3**
Is sea surface temperature masked during the times of ice cover? It think that it does not as it
is a boundary condition defining permafrost temperature in the nearshore and the
shore/shoreface.

Yes, the statement this comment refers to is correct, and the sea surface temperature is indeed
masked during periods of sea ice cover.  This is not a problem in terms of boundary

conditions, because the model is simply not activated during times of ice cover (see added flow chart in response to Reviewer #2, which is a new Figure 1 in the revised manuscript). We have assumed negligible erosion takes place when the coast is covered in sea ice (although this has the potential to be developed in future work).

**Commented [AS5]:** Please Comm2 One may suggest to still outline the morphologies where the model could be applicable.

We thank Anatoly for this comment, and also the associated comments throughout the manuscript where this comes up. We have added a statement to the abstract referring to lithified ice-poor coastlines (see our response above to the first bullet point of the 'Main points.'

**Commented [AS6]:** Please see Coom1

Yes, please see our response to Comm1 and also Comm2. In addition, we have changed the the wording 'partially frozen coastlines' in this sentence also to 'permafrost coasts'.

**Commented [AS7]:** See Coom2

We have changed a statement in the abstract to mention lithified coastlines, please see our response to Comm2 and also Comm1.

**Commented [AS8]:** See Comm2. I again would like to mention that there are also other morphologies in the Arctic. More sever wave climate will for sure lead to stronger erosion on sandy beaches. And sandy beaches, at least in some cases do not have frozen cliff in summer due to deep active layer.

Yes, we thank Anatoly for this comment that we should include more statements referring to other types of coastlines. We have reworded the first sentence in the introduction so that it now reads:

Section 1, Lines 26-29: 'Due to warmer temperatures and reduced sea ice protection from bigger waves (Casas-Prat and Wang, 2020; Overeem et al., 2011), especially as freeze-up becomes delayed further into the fall storm season, Arctic coastlines are becoming increasingly vulnerable to the erosion of sandy beaches and destabilization of permafrost cliffs (Sinitsyn et al., 2020; Biskaborn et al., 2019).'

Page 2.

**Commented [AS9]:** Comm6. What about the roles of coastal types? Why it is not mentioned when it comes to the variability of erosion rates?

Yes, this is a good point and we have added the word 'ice-rich' in this statement to clarify which types of coastlines we are referring to (Section 1, Line 30).

**Commented [AS10]:** Comm7. It sounds somewhat confusion to mask different geomorphologies by referring to locations. See Comm2

Yes, we have replaced the word 'some locations' with 'some geomorphologies' in this statement (Section 1, Line 42). Please also see our response to the first bullet of the main points/Comm2.

**Commented [AS11]:** Why the authors do not mention typical coastal types, and corresponding coastal processes, which would be thermal abrasion and thermal denudation when it comes to unlithified ice-rich coasts?

We have now made several references to other types of coastlines in the text. For these, please see our responses to the first and second bullet point of the reviewer's main comments above, as well as comment 'AS8' and Comm2. In these responses, we have added new references to the processes of thermal abrasion and thermal denudation, as well as references to ice-poor, sandy or lithified coastlines.

**Commented [AS12]:** Comm9. Were those villages placed on an ice-rich coast? Probably some of them were.

Yes, that is indeed correct.

**Commented [AS13]:** Comm15. When it comes to the motivation behind this article and the biogeochemistry, would be good to refer to Lantuit et al. (2012) and point out that model can be helpful for moving from a static definition of organic carbon (as Lantuit et al. did) to a dynamic.

Yes, we thank Anatoly Sinitsyn for this comment and have now added the following statement where the biogeochemical modelling is mentioned:

Section 4.3, Lines 471-472: 'Such dynamic estimation of nearshore biogeochemistry would be an improvement to using estimates of coastline retreat and static coastal carbon content (Lantuit et al., 2012; Wegner et al., 2015).

The Arctic Coastal Dynamics Database: A New Classification Scheme and Statistics on Arctic Permafrost Coastlines

Page 4.

**Commented [AS14]:** Comm. Would be good to mention that conduction is not taken into account.

Yes, agreed. We have added the following statement also in response to the comments in the Methods section from Reviewer #2:

Section 2.1.2, Lines 159-163: "Consistent with the chosen erosion module in ArcticBeach v1.0, Kobayashi et al. (1999), conductive heat transfer and solar radiation are not directly included. Solar radiation can be partially accounted for in the sea surface temperature input and sea ice cover (see Section 2.3). Conduction effects are much smaller than effects of solar radiation over long time periods and are neglected. However, the opportunity to include effects of solar radiation can be implemented in later versions of the model, to include processes such as thaw slumping and 1-D heat-transfer permafrost models as described in Section 4.2.1."

**Commented [AS15]:** Comm12. Sediments in the coastal zone in the Arctic are normally saline, hence freezing temperature is lower that 0 C.

Yes, we are aware that salinity does impact the freezing temperature, but we simplified the study such that 0°C was used and did not take into account salinity measurements. However, we realize it was not clear in the manuscript that this was an assumption, and have now added the following to Section 2.1.1, Lines 97-98:
'…(assumed in this study to be 0°C, but can also be adjusted using salinity data near the coastline).'

**Commented [AS16]:** Comm13. What is Tm?

Yes, thank you, we missed a label for $T_m$ and have now added the following reference:

Section 2.1.1, Lines 105-106: '… and $T_m$ [°C] is the thawing temperature of the frozen sediment.'

**Commented [AS17]:** Comm14. "heat transfer coefficient" of the sea water?

We have now added an equation (Eq. 2) that describes the heat transfer coefficient. Please see our response to the second bullet point in the main comments above.

Page 5.

**Commented [AS18]:** Comm16. Long/shore sediment transport also defines erosion at the site when it comes to clastic sediments (sandy beaches). It would be useful just to mention that long-shore sediment transport is not considered by the model

Yes, we have now added the following statement:

Section 2.1.2, Lines 130-131: 'Long-shore transport also defines erosion on sandy beaches but is currently neglected in this 1-D approach.'

**Commented [AS19]:** Comm18. What does this parameter mean? Please defied this parameter.

We have added the following description of 'sand roughness length':

Section 2.1.4, Line 185: '…(assumed to be 2.5 times the median sediment diameter (Nielsen, 1992)) ...'

Page 7.

**Commented [AS20]:** Comm19. 10 m/s ? What defines the selected value of 10 m/s/?
It is known that lower wind speeds are also capable to generate a storm. It is Ok to use 10 m/s, but one should then outline that this is somewhat a characteristic value. Jus [supplement pdf is cut off here so we cannot see the rest of the comment].

No, the manuscript states: '10m east and west wind speed vectors' and not '10 m/s'. This refers to wind vectors from reanalysis data that has been taken at 10m height. So, this means we include all wind speeds and do not have a threshold.

Page 9.

**Commented [AS21]:** Comm20. It would be good to clarify then that, in fact, some of the winter storms might have been taken into account (which brings us closer to the reality).

Yes, this is a good point and we have added the following statement here:

Section 2.3, Lines 224-225:  'Winter storms can occur over less than 15% sea ice cover, so when this happens, erosion is still simulated during winter.'